# Modulation of AggR levels reveals features of virulence regulation in enteroaggregative *E. coli*

Alejandro Prieto [1], Manuel Bernabeu[1], José Francisco Sánchez-Herrero [2], Anna Pérez-Bosque [3,4], Lluïsa Miró[3,4], Christine Bäuerl[5], Carmen Collado [5], Mário Hüttener [1✉] & Antonio Juárez [1,6✉]

Enteroaggregative *Escherichia coli* (EAEC) strains are one of the diarrheagenic pathotypes. EAEC strains harbor a virulence plasmid (pAA2) that encodes, among other virulence determinants, the *aggR* gene. The expression of the AggR protein leads to the expression of several virulence determinants in both plasmids and chromosomes. In this work, we describe a novel mechanism that influences AggR expression. Because of the absence of a Rho-independent terminator in the 3′UTR, *aggR* transcripts extend far beyond the *aggR* ORF. These transcripts are prone to PNPase-mediated degradation. Structural alterations in the 3′UTR result in increased *aggR* transcript stability, leading to increased AggR levels. We therefore investigated the effect of increased AggR levels on EAEC virulence. Upon finding the previously described AggR-dependent virulence factors, we detected novel AggR-regulated genes that may play relevant roles in EAEC virulence. Mutants exhibiting high AggR levels because of structural alterations in the *aggR* 3′UTR show increased mobility and increased pAA2 conjugation frequency. Furthermore, among the genes exhibiting increased fold change values, we could identify those of metabolic pathways that promote increased degradation of arginine, fatty acids and gamma-aminobutyric acid (GABA), respectively. In this paper, we discuss how the AggR-dependent increase in specific metabolic pathways activity may contribute to EAEC virulence.

[1] Department of Genetics, Microbiology and Statistics, Universitat de Barcelona, Barcelona, Spain. [2] High Throughput Genomics and Bioinformatics Facility, Institut Germans Trias i Pujol, Badalona, Spain. [3] Department of Biochemistry and Physiology, Universitat de Barcelona, Barcelona, Spain. [4] Institut de Nutrició i Seguretat Alimentària, Universitat de Barcelona, Barcelona, Spain. [5] Institute of Agrochemistry and Food Technology, National Research Council (IATA-CSIC), Paterna, Valencia, Spain. [6] Institute for Bioengineering of Catalonia, The Barcelona Institute of Science and Technology, Barcelona, Spain. ✉email: mhuttener@me.com; ajuarez@ub.edu

Enteroaggregative *Escherichia coli* (EAEC) are among the diarrheagenic *E. coli* pathotypes[1]. EAEC are the focus of active research because of their roles as etiologic agents of enteric infections that cause acute and persistent diarrhea in children and adults[2–6], persistent diarrhea in HIV-infected patients, traveler's diarrhea and extraintestinal infections such as urinary tract infections[7–9]. EAEC strains are genetically heterogeneous in nature. The etiologic agent of an outbreak of foodborne hemorrhagic colitis in Germany in 2011 was an EAEC strain that displayed specific genomic features[10]. It was lysogenized with a phage that encodes the Stx2a toxin. The identified clone, which belongs to the O104:H4 serotype, expresses a wide variety of virulence factors in addition to the Stx2a toxin, including several other typical EAEC virulence factors. A consequence of EAEC genetic heterogeneity is that many virulence determinants are not present in all strains[5,11,12], and the mechanisms by which EAEC cause disease are still poorly understood[13]. A common genomic feature of most EAEC strains is the presence of a virulence plasmid termed pAA. pAA plasmids encode a transcriptional activator named AggR, which is a member of the AraC-XylS family of transcription factors[14]. AggR expression is sensitive to several factors. Negative regulators of *aggR* expression include the nucleoid-associated protein H-NS[15] and the pAA plasmid-encoded member of the AraC negative family of regulators, the Aar protein[16,17]. Both the FIS protein and the nucleotide second messenger (p)ppGpp are required for AggR expression[18,19]. AggR also autoactivates its own transcription[15].

Because of its central role in EAEC virulence, the AggR regulon has been thoroughly studied, mainly in the prototypical EAEC strain 042; this strain caused diarrhea in a volunteer trial[20], and its genome sequence has been available for several years[21]. Strain 042 harbors the IncFIIA virulence plasmid pAA2[21,22], which encodes, among other virulence determinants, the AggR virulence regulator and the AAF/II variant of a fimbrial adhesion determinant required for adherence to human intestinal mucosa[21,23]. The genes subjected to AggR regulation were initially identified in EAEC strain 042 through the use of microarray technology[24]. At least 44 genes were found to be regulated by AggR and are localized both in the pAA2 plasmid and in the bacterial chromosome. AggR-dependent plasmid-encoded virulence factors include aggregative adherence fimbriae (AAF), of which there exist different variants[23,25–28], the Aap dispersin and its type I secretion system (TISS), the polysaccharide deacetylase encoded by the *shf* gene, and the *Shigella flexneri* virulence protein VirK[13,24,29,30]. The chromosomal virulence determinants regulated by AggR, including a type VI secretion system identified in strain 042[24], map mainly to chromosomal islands. A recent study used RNA-seq technology to further explore the AggR regulon in strain 042[13]. When comparing the transcriptome of the wild-type 042 strain with that of an isogenic *aggR* mutant, as many as 112 genes were found to be differentially expressed. All the AggR-regulated genes except one identified in a previous report[24] were identified in the latter study. Eighty-three AggR-regulated genes were found to be located in the chromosome. They include the Aai type VI secretion system and several others of unknown function. During our continuous research on *aggR* regulation[19], we found an unexpected result. The *aggR::lacZ* transcriptional fusion generated by placing the *lacZ* gene after the *aggR* ORF in the 3′UTR (thus enabling *aggR* promoter-dependent β-galactosidase expression in the presence of wt levels of AggR) produced clones that showed a phenotype different from that of the *E. coli* 042 wt strain (i.e., it had a different colony morphology, different growth rate, and different degree of cell adherence). The study of this and similar clones has led to the conclusion that sequences in addition to those in the *aggR* ORF influence the expression of AggR at the posttranscriptional level. Disruption of these sequences results in increased AggR expression, which leads to the induction of the AggR regulon. In turn, by studying the effect of induced AggR expression on *E. coli* 042 virulence, we expand our knowledge of the AggR regulon and suggest novel mechanisms by which EAEC may cause disease.

## Results

**Clones of strain 042 that harbor an *aggR::lacZ* transcriptional fusion after the last *aggR* coding codon show enhanced cell aggregation and reduced growth rates.** AggR activates its own transcription[15]. The use of *aggR::lacZ* transcriptional fusion for regulatory studies may require the placement of the *lacZ* cassette after the *aggR* coding sequence, thus retaining AggR function. These transcriptional fusions show higher β-galactosidase than fusions that contain the *lacZ* reporter gene within the *aggR* coding sequence (Fig. 1a). Unexpectedly, all the obtained derivatives from strain 042 Δ*lacZ* containing the *lacZ* gene immediately after the *aggR* coding sequence (strain 042 *aggR* + *lacZ*3′UTR) showed a different phenotype than that of the parental 042 Δ*lacZ* strain both on agar plates and in liquid medium. The colonies formed by these 042 *aggR* + *lacZ*3′UTR derivatives are rougher than those formed by the 042 Δ*lacZ* strain, and when cultured in liquid medium, they have a much faster sedimentation rate than the 042 Δ*lacZ* strain. They also showed a reduced growth rate (Supplementary Fig. 1). These phenotypes were also observed for the clones generated before the *lacZ*-Km^r cassette was inserted, that is, the clone harboring the FRT sequence following the *aggR* stop codon (strain 042 *aggR* + FRT3′UTR). To rule out polar effects of the insertion of the *lacZ* gene and its associated Km^r cassette as a cause of the observed phenotype, we used strain 042 *aggR* + FRT3′UTR for further studies.

Taking into account that sedimentation can be a consequence of cellular aggregation, we decided to compare the cell aggregation of both 042 wt and 042 *aggR* + FRT3′UTR strains. We also used strain 042 Δ*aggR* and a plasmid-free derivative of strain 042 (strain 042 pAA2-) for the study. The results are shown in Fig. 1b, c. Compared with the wt 042 strain, strain 042 *aggR* + FRT3′UTR showed increased cell aggregation at 37 °C. The fact that neither strain 042 Δ*aggR* nor strain 042 pAA2- aggregated at any temperature is consistent with the observation that cell aggregation depends on the expression of AggR-activated plasmid genes. The effect is temperature-dependent, as has been reported for the expression of the AggR regulon[19]. In turn, the expression of AggR itself is temperature-regulated (Supplementary Fig. 2).

**Insertion of an FRT sequence in the 3′UTR of the *aggR* gene of strain 042 results in the increased expression of the AggR-regulated *aafA* and *aap* genes.** To further characterize the effect of the alteration of *aggR* 3′UTR sequences on cell physiology, we analyzed whole-cell extracts of strains 042 wt, 042 Δ*aggR*, 042 pAA2- and 042 *aggR* + FRT3′UTR with SDS-PAGE (Fig. 2a). An overexpressed protein of low molecular mass was identified in the total cell extract of strain 042 *aggR* + FRT3′UTR. We then obtained the cytoplasmic, periplasmic, inner- and outer membrane cellular fractions of these strains and analyzed them with SDS-PAGE. The overexpressed protein in strain 042 *aggR* + FRT3′UTR was located in the outer membrane fraction (Fig. 2b). To identify the protein, the band was excised from the acrylamide gel and analyzed by LC-MS/MS. It corresponds to the major subunit of aggregative adherence fimbriae type II (AafA). We then decided to analyze the cell-free secreted proteins of all four strains with SDS-PAGE (Fig. 2c). Strain 042 secretes two proteins of large molecular mass (104 and 116 kDa) into the extracellular medium. The 104 kDa protein corresponds to the product of the

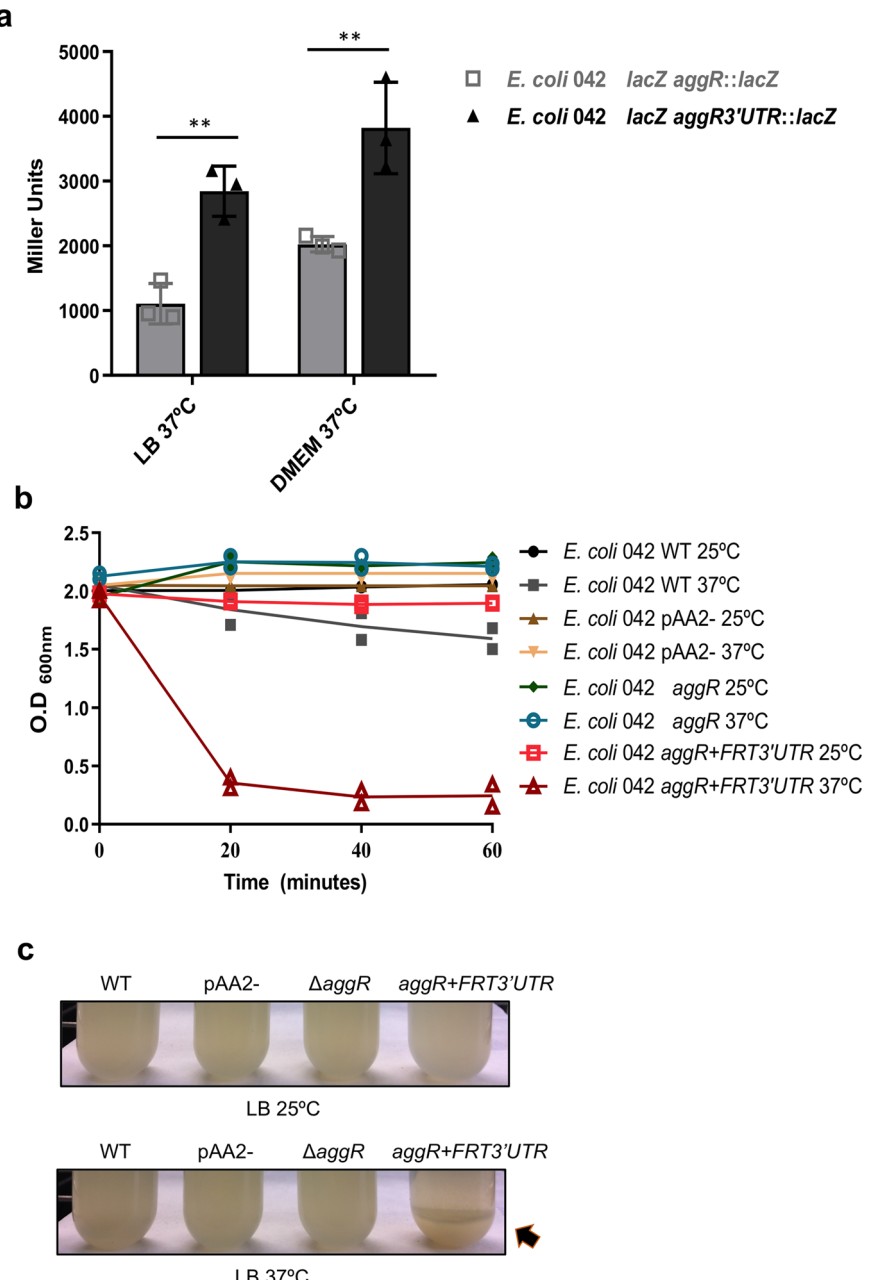

**Fig. 1 Expression of transcriptional *aggR::lacZ* fusions and quantification of cellular aggregation of *E. coli* strain 042 and derivatives. a** The *aggR::lacZ* and *aggR3´UTR::lacZ* strains were grown in LB medium or DMEM at 37 °C. Samples were collected at the onset of the stationary phase (OD$_{600}$ 2.0). The *aggR::lacZ* construct contains the *lacZ* gene in the coding region of the *aggR* gene, and the *aggR3'UTR::lacZ* construct contains the *lacZ* gene reporter fusion located after the stop codon of the *aggR* gene. The results show the means and the standard deviation of three independent experiments. Significance was analyzed by unpaired two–sided Student's t-test. Significance is indicated as **$p < 0.01$. **b** Quantification of cellular aggregates as indicated by a decrease in the OD$_{600}$. *E. coli* 042 wt, pAA2-, Δ*aggR* and *aggR* + *FRT3'UTR* cell aggregates at 25 °C and 37 °C in LB medium. **c** Picture of a representative experiment showing the hypercellular aggregation of the *aggR* + *FRT3'UTR* strain at 37 °C (black arrow).

plasmid-encoded *pet* gene, an autotransporter with enterotoxin activity[31]. This protein was detected in all the strains analyzed except the 042 pAA2- strain. The second protein of large molecular mass corresponds to the Pic protein, a chromosome-encoded serine protease[32]. Pic was detected in all four strains. In addition, a protein of low molecular mass was overexpressed in strain 042 *aggR* + *FRT3'UTR* with respect to the wt strain, but it was not expressed in strains 042 Δ*aggR* or 042 pAA2-. The analysis of the excised band by LC-MS/MS showed that it corresponds to Aap dispersin. The *aap* gene is plasmid-encoded and AggR-regulated[29].

**Northern blot analysis of the *aggR* transcript.** All the specific phenotypes of strain 042 *aggR* + *FRT3'UTR* (i.e., increased cell aggregation and increased expression of the AafA and Aap proteins) strongly suggest that insertions of DNA sequences into the 3'UTR of the *aggR* gene resulted in increased AggR expression. To confirm this hypothesis, we analyzed the transcription of the *aggR* gene by Northern blotting. Strains 042 wt, 042 Δ*aggR* and 042 *aggR* + *FRT3'UTR* were used for the analysis. Three different DNA probes were used. Two of these probes, designed to be complementary to the sense strand, mapped within the 3' region of the *aggR* coding sequence (Supplementary Fig. 3). To detect

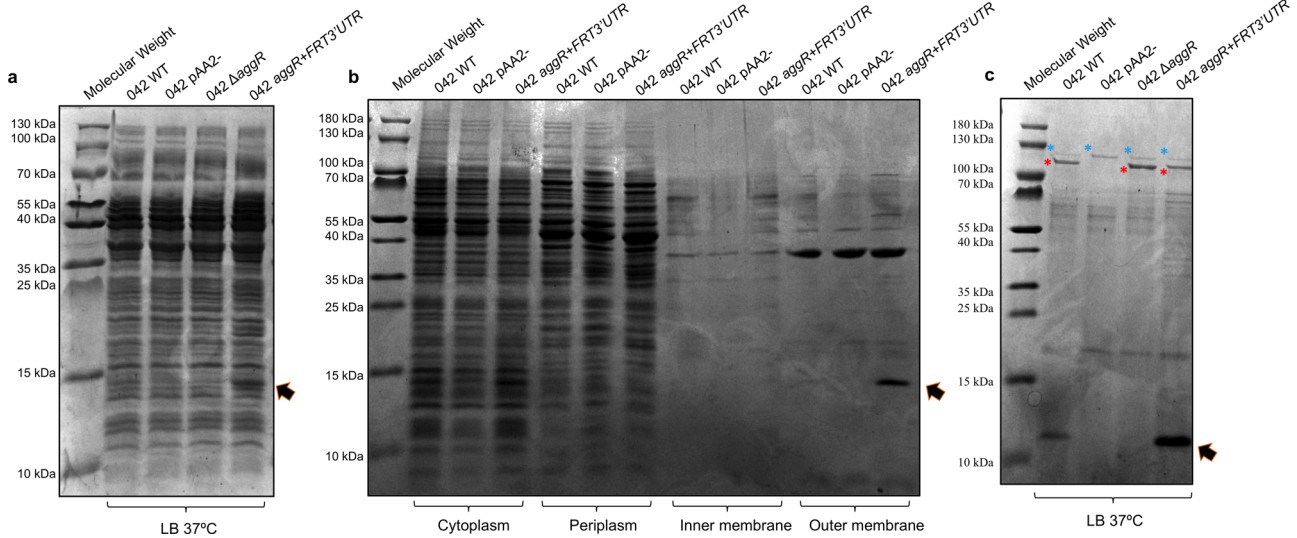

**Fig. 2 The AafA and Aap proteins are overexpressed in the *aggR* + *FRT3′UTR* strain.** SDS-PAGE analysis of (**a**) whole cell extracts, (**b**) cellular fractions, and (**c**) the cell-free secreted proteins of the *E. coli* O42 wt, pAA2-, Δ*aggR* and *aggR* + *FRT3′UTR* strains. Black arrows in (**a**) and (**b**) point to the overexpressed AafA protein in the a*ggR* + *FRT3′UTR* genetic background. Blue asterisk in (**c**) correspond to the Pic protein, and red asterisk highlight the Pet protein, respectively. The black arrow in (**c**) points to the overexpressed dispersin protein in the *aggR* + *FRT3′UTR* genetic background.

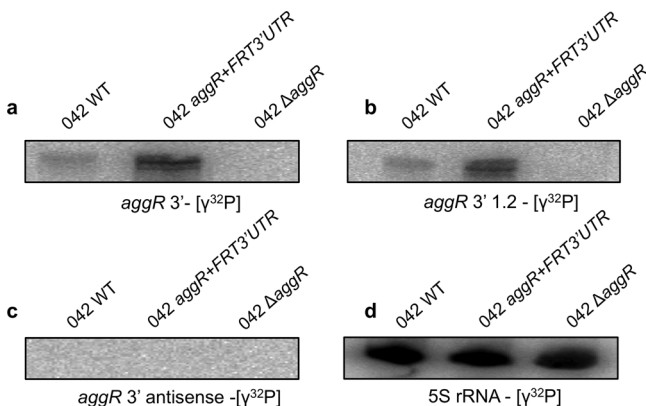

**Fig. 3 Northern blot analysis of the *aggR* transcript.** RNA was extracted from strains *E. coli* O42 wt and *aggR* + *FRT3′UTR*, with the Δ*aggR* strain used as a control for the specificity of the probes. **a** and **b** correspond to the levels of the *aggR* transcript detected by [γ³²P] probes that were complementary to the sense strands, while (**c**) corresponds to the levels of the *aggR* transcript detected using a [γ³²P] probe complementary to the antisense strand of the *aggR* gene. **d** 5 S rRNA was used as the loading control. The experiment was repeated three times. A representative experiment is presented.

antisense transcripts within the *aggR* region, the third probe was designed to be complementary to the antisense strand (Fig. 3 and Supplementary Fig. 3). An aggR-specific transcript was detected by using either probe hybridizing the sense transcript. No antisense transcript was detected. As expected, the levels of the *aggR* transcript were higher in strain O42 *aggR* + *FRT3′UTR* than in the wt strain. Unexpectedly, the *aggR* transcript detected (greater than 1.3 kb) was significantly larger than that of the *aggR* gene (800 bp) (Supplementary Fig. 3).

**Bioinformatic analysis of the 3′UTR and downstream sequences of the *aggR* gene in pAA plasmids.** The fact that insertions of foreign DNA downstream the *aggR* gene result in increased transcript levels and that the size of the *aggR* transcript is unexpectedly large prompted us to perform a detailed analysis

of the DNA sequences located 3′ downstream of the *aggR* gene in the pAA2 plasmid and in the pAA plasmids of strains C700-09, 55989 and O104:H4. Strain C700-09 expresses two different aggregative fimbriae (AAF/III and AAF/V)[33]. Strain 55989 is the prototypic EAEC strain expressing aggregative fimbriae type III (AAF/III)[27]. Strain O104:H4 is the etiologic agent of the SUH outbreak in Germany in 2011 and expresses AAF/I fimbriae. In the pAA2 plasmid, an IS*1A* element and the *aafDA* operon are located downstream of the *aggR* gene (Fig. 4a). When the genomic structure of the region where the *aggR* gene is located in the pAA2 plasmid is compared with the corresponding region in the pAA plasmids of the other EAEC strains, relevant information can be obtained (Fig. 4b). Both the promoter and coding sequences of the four compared *aggR* genes share more than 97% homology. A high degree of homology is also shared by a short (44 bp) nucleotide stretch located 3′ downstream of the *aggR* translational stop codon. Hence, we think that the common region is the 3′UTR of the *aggR* gene (Fig. 4c). In all four 3′ downstream regions, IS elements are located downstream of the corresponding *aggR* gene. All the identified IS elements, out of the IS*1A* element of the pAA2 plasmid share a high degree of homology. Specifically considering the 3′ downstream region of the *aggR* gene in plasmid pAA2 (Fig. 4c), the IS*1A* element is located seven nucleotides downstream of the 3′UTR sequence of the *aggR* gene. It includes two complete 50 nucleotide-long inverted repeats (IRL and IRR) and has a total length of 768 nucleotides.

Taking into account that the detected *aggR* transcript is unexpectedly long, we also used an *in silico* approach (see the materials and methods section for details) to detect both Rho-independent and Rho-dependent transcriptional terminators downstream of the *aggR* gene (Fig. 4d). Interestingly, in the downstream sequences of the pAA2-encoded *aggR* gene, the first Rho-independent transcriptional terminator was detected approximately 3.7 kb downstream of *aggR*. In contrast, several potential Rho-dependent terminators were found, with the first two provided by the IS*1A* element. A similar situation was found in the other downstream sequences of the *aggR* genes from pAA plasmids from EAEC strains C700-09, 55989 and O104:H4. Rho-independent terminators are located 550 bp and 1500 bp from the end of the *aggR* coding sequence of strains C700-09 and 55989,

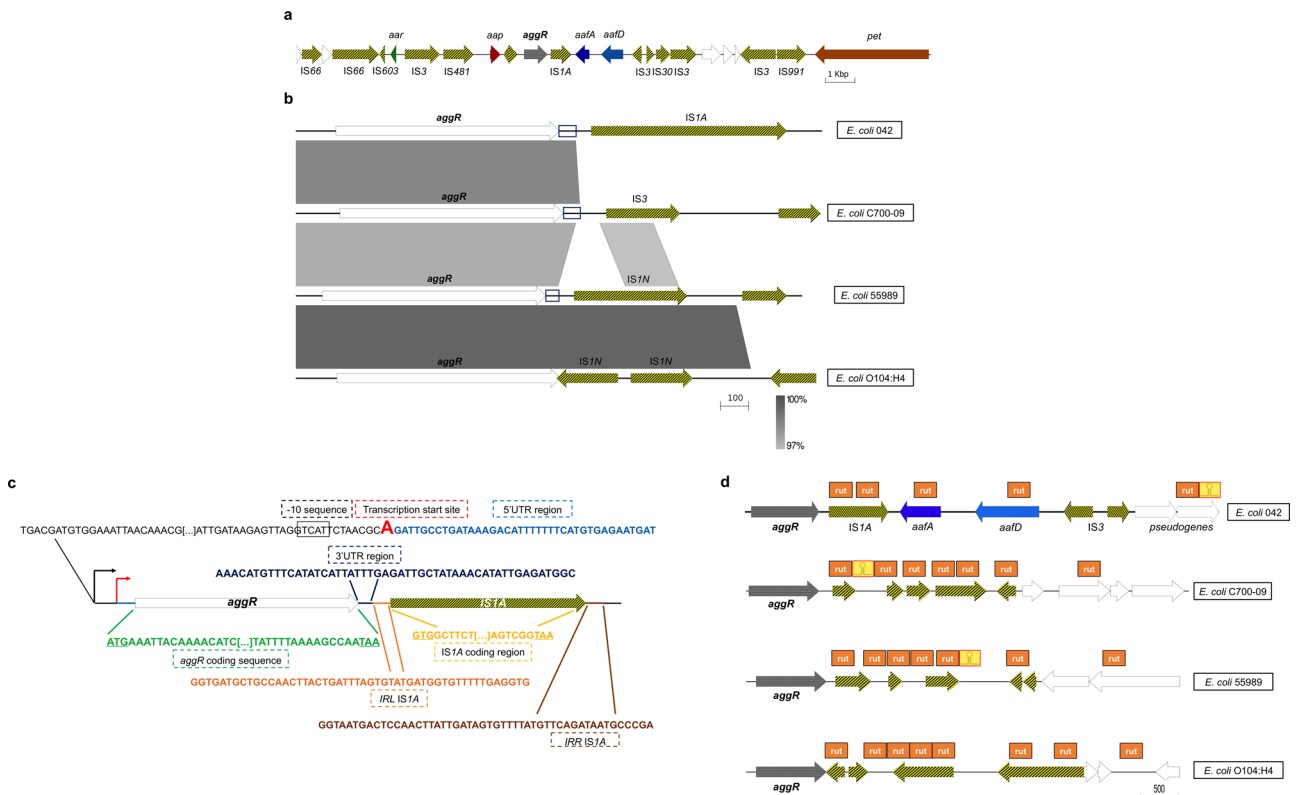

**Fig. 4 Detailed map of the *aggR* genetic region and *in silico* prediction of Rho-dependent and Rho-independent terminators that map downstream of the *aggR* gene coding sequences. a** Details of the genetic region to which the *aggR* gene maps. Genetic determinants flanking the *aggR* gene of the pAA2 plasmid in strain 042. Insertion elements are indicated in black and yellow. Arrows indicate the direction of transcription. To simplify the interpretation, the reverse complement was used to show the *aggR* gene in the sense strand. **b** Alignment of the regions that include the *aggR* gene and downstream sequences in the pAA plasmids of strains 042, C700-09, 55989 and O104:H4, respectively. **c** Detailed map of the *aggR* gene and 3′UTR sequences in plasmid pAA2. Black letters correspond to the 3′ end of the promoter sequence of the *aggR* gene. The 5′UTR (light blue letters) and 3′UTRs (dark blue letters) of *aggR* are shown, as are the sequences corresponding to the IS*1A* element located downstream of the gene (orange letters, IRL elements; yellow letters, IS*1A* coding region; and brown letters, IRR element). The -10 element of the *aggR* promoter as well as the *aggR* transcription start site are also shown. The black arrow corresponds to the start of the promoter region, and the red arrow corresponds to the *aggR* transcriptional start site. **d** Diagram showing the *in silico* prediction of Rho-dependent and Rho-independent terminators that map downstream of the *aggR* gene coding sequences. Rho-independent (yellow) and Rho-dependent (rut, in orange) terminators (in silico predicted, see material and methods) located downstream of the *aggR* coding sequences in the pAA plasmids of *E. coli* strains 042, C700-09, 55989 and O104:H4 are shown.

respectively, whereas no Rho-independent terminator was found in the proximity of the *aggR* gene in *E. coli* strain O104:H4. In contrast, Rho-dependent terminators are located close to the 3′ downstream sequences of the *aggR* genes encoded in all three pAA plasmids (Fig. 4d).

**Clones containing an insertion of an FRT sequence in the 3′ UTR of the *aggR* gene of strain 55989 show a phenotype similar to that of strain 042 *aggR*+FRT3′UTR.** The observed sequence similarities in the 3′UTR regions of the different EAEC strains analyzed suggest that they may play regulatory roles similar to those found in strain 042. To support this hypothesis, we inserted the FRT sequence following the *aggR* stop codon in the 3′UTR region of the *aggR* gene of the EAEC strain 55989[27]. Its genome shows a very close relationship with that of the EHEC O104:H4 outbreak strain but express a different AAF fimbrial type. A 55989 *aggR* + FRT3′UTR clone was compared with the wt 55989 strain for cellular aggregation and for growth kinetics. When compared to the wt strain, strain 55989 *aggR* + FRT3′UTR showed enhanced cellular aggregation and a reduced growth rate (Fig. 5). In addition, the major subunit of the AAF/III determinant (Agg3A and the dispersin (Aap) proteins) was overexpressed (Fig. 5).

**Deletion of different DNA fragments downstream of the *aggR* gene of strain 042 results in increased levels of the *aggR* transcript.** Upon obtaining detailed information on the structure of the 3′ downstream region of the *aggR* gene in plasmid pAA2, we deleted different sequences to test whether they play a role in the posttranscriptional regulation of *aggR* expression. The first deletion corresponded to the 44 bp 3′UTR established in silico. The second corresponded to the IRL sequence of the IS*1A* element. The third included both regions, and the last one corresponded to the complete deletion of the IS*1A* element. *aggR* transcription was assessed in all four constructs by Northern blotting (Fig. 6). The results showed that all the deletions had the same effect: they significantly increased *aggR* transcript levels.

***aggR* transcripts extend beyond the *aggR* gene.** The unexpectedly large *aggR* transcript, the absence of Rho-independent transcriptional terminators downstream of the *aggR* ORF and the fact that IS*1A* sequences influence the levels of the *aggR* transcript suggest that the transcripts initiated at the *aggR* promoter extend far beyond the *aggR* ORF. We used walking RT-PCR to verify this supposition. RNA was extracted from the LB cultures of strains 042 wt and 042 *aggR* + FRT3′UTR grown to the onset of the stationary phase (OD$_{600}$ of 2.0) and retrotranscribed to cDNA.

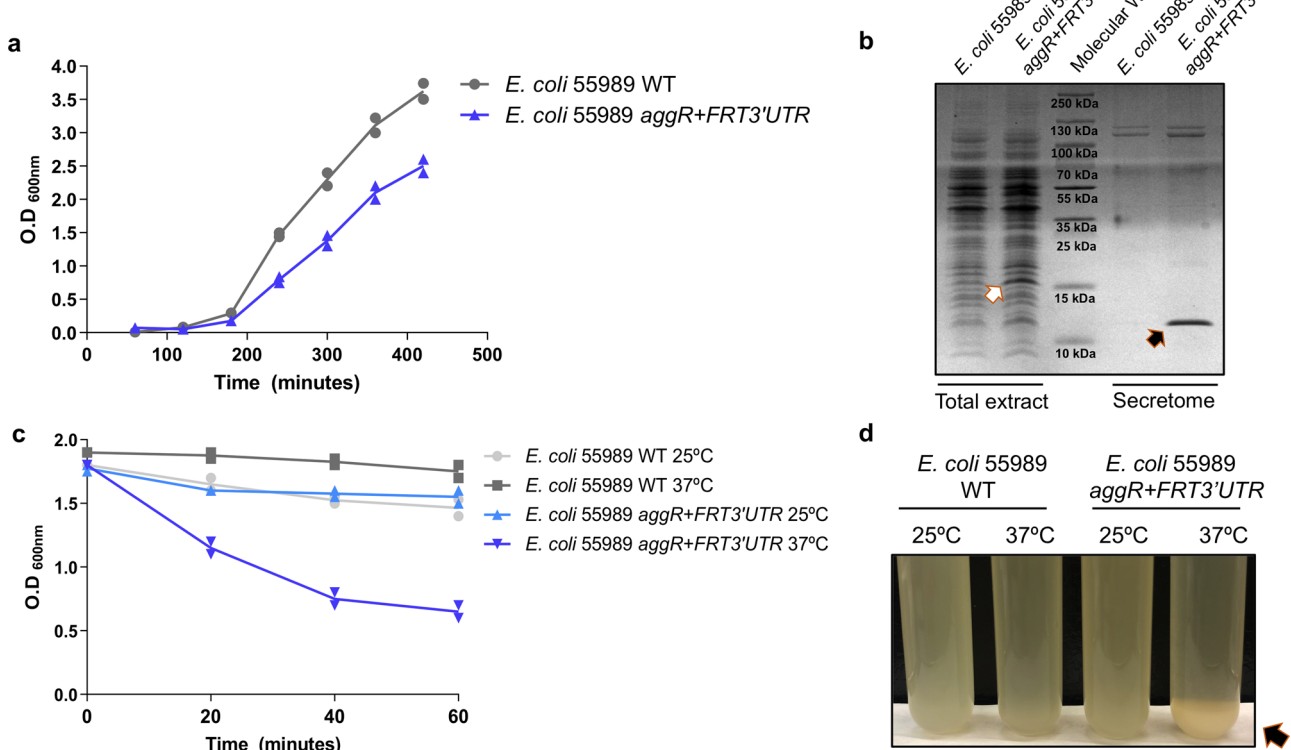

**Fig. 5 Alteration of the 3′UTR gene of *E. coli* 55989 enteroaggregative strain shows the same phenotype as the *E. coli* O42 strain. a** Growth curves of *E. coli* strains 55989 wt and *aggR* + *FRT3′UTR* in LB medium. **b** SDS-PAGE analysis of whole cell extracts (left) and the cell-free secreted proteins (right) of the *E. coli* 55989 wt and *aggR* + *FRT3′UTR* strains. White arrow points to the overexpressed Agg3A protein, while black arrow points to the Aap protein overexpressed in the secretome of the strain *E. coli* 55989 *aggR* + *FRT3′UTR*. **c** Quantification of cellular aggregation as indicated by a decrease in the $OD_{600}$. **d** Picture of a representative experiment showing the hypercellular aggregation of the 55989 *aggR* + *FRT3′UTR* strain at 37 °C (black arrow).

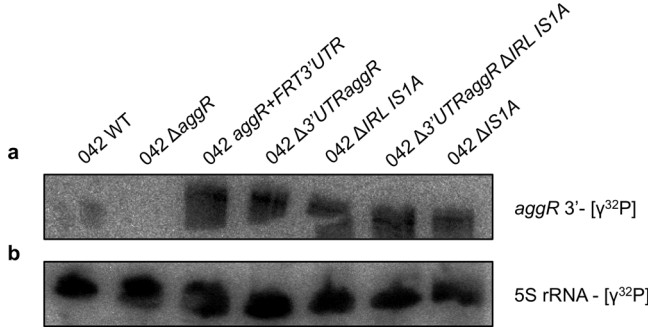

**Fig. 6 Northern blot analysis of the O42 *aggR* transcript in different genetic backgrounds harboring different deletions in the 3′ sequences of the *aggR* gene. a** mRNA levels of the *aggR* gene detected by using a [γ32P] probe complementary to the end of the *aggR* gene sense strand of different strains. **b** The 5 S rRNA used as a loading control. The experiment was repeated three times, and a representative experiment is shown.

The resulting cDNA was used for different amplified PCRs by using a fixed forward oligonucleotide located at the 3′ end of the *aggR* coding sequence and different reverse oligonucleotides complementary to different sequences located 3′ downstream of the *aggR* gene (Fig. 7a–g). In all the amplification reactions, the amount of *aggR* cDNA detected was higher in the O42 *aggR* + *FRT3′UTR* strain than in the O42 wt strain (Fig. 7b–g). With respect to the former strain, the amplification products corresponding to the *aggR* transcript were detected with reverse oligonucleotides complementary to the IS*1A* element, thus confirming the existence of *aggR* transcripts that, when

extended beyond the *aggR* gene, reach, at least, the IS*1A* element (Fig. 7f).

**PNPase participates in the degradation of the *aggR* transcript**. Previous reports have shown that RNaseE and PNPase can participate in the degradation of 3′UTR-mediated mRNA[34–37]. We therefore sought to determine whether any of these RNases plays a role in the decay of the *aggR* transcript in strain O42. To that end, we constructed O42 derivatives lacking RNase E or PNPase function. Considering that the N-terminal region of RNase E is essential for *E. coli* growth, we constructed RNase E mutants lacking only the C-terminal region of the protein[38]. Mutants lacking the PNPase function were constructed by deleting the corresponding gene. *aggR* transcription was analyzed by Northern blot with strain O42 wt and the corresponding derivatives lacking either RNaseE or PNPase function. The *aggR* transcription levels of the *aggR* gene were low in the wt strain and the RNAse E mutant, and high in the PNPase mutant and in the strain O42 *aggR* + *FRT3′UTR* (Fig. 8). The bacterial cells carrying the PNPase mutation also showed the same phenotype of high cellular aggregation as displayed by strain O42 *aggR* + *FRT3′UTR* (Fig. 8).

**The effect of increased *aggR* transcript levels on the global transcriptome of strain O42**. To characterize the *aggR* regulon, previous studies compared the transcriptome of the wt O42 strain with that of an *aggR* isogenic mutant lacking the AggR protein[13,24]. Taking into account that strain O42 *aggR* + *FRT3′UTR* expresses higher AggR levels than the wt O42 strain, we decided to use both strains to obtain information on the AggR

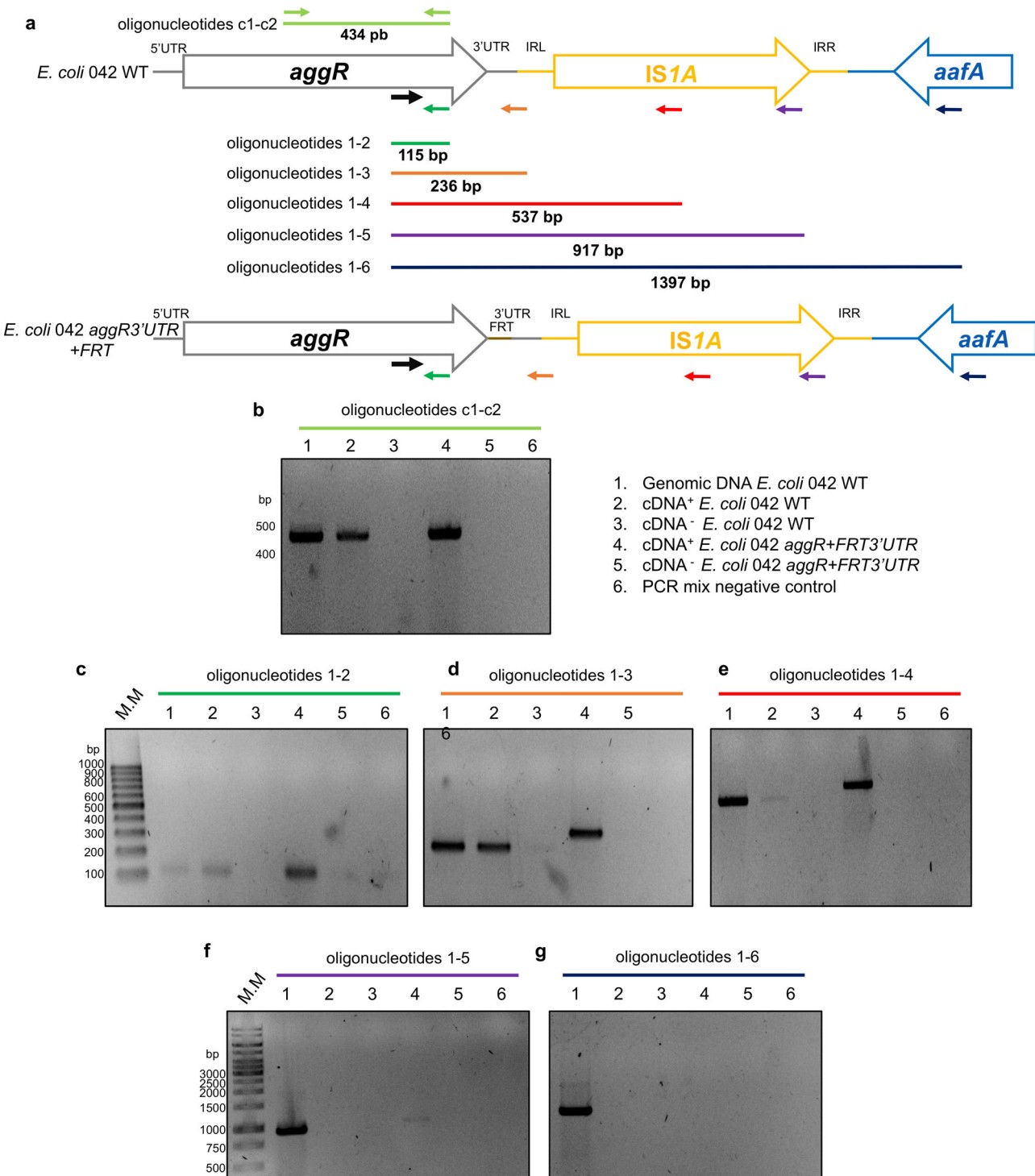

**Fig. 7 Walking RT-PCR performed to analyze the length of the *aggR* transcript. a** Details of the oligonucleotides used in the walking RT-PCR experiment and the lengths of the corresponding predicted amplicons. **b–g** Amplification products corresponding to the *aggR* transcript detected with the different sets of primers used. The experiment was repeated three times, and a representative experiment is shown.

regulon that might be complementary to that obtained in strains with an *aggR* genetic background. These strains were grown in LB medium at 37 °C until the onset of the stationary phase ($OD_{600}$ 2.0). RNA was then isolated, and the transcriptome of both strains was obtained by RNA-seq. All the previously reported genes that had been identified in previous works as down-regulated in *aggR* mutants[13,24] were accordingly upregulated in strain 042 *aggR* + *FRT3′UTR* (Supplementary Table 1).

Supplementary Table 2 shows the 45 genes exhibiting the highest fold change values in strain 042 *aggR* + *FRT3′UTR* compared to their expression in the wt strain. In addition to genes coding hypothetical proteins, we could identify several metabolic genes which were upregulated. Almost all of these genes belong to three metabolic pathways: degradation of arginine, degradation of fatty acids and degradation of γ-aminobutyric acid (GABA) (Supplementary Fig. 4a, b). In addition, we identified upregulated genes

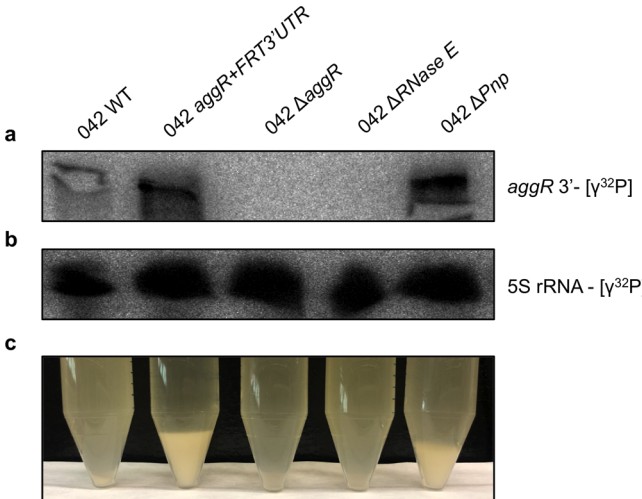

**Fig. 8 Northern blot analysis of the *aggR* transcript of *E. coli* strains O42 wt, *aggR* + *FRT3′UTR*, Δ*aggR* and the O42 derivatives lacking either RNaseE or PNP ribonuclease function. a** mRNA levels of the *aggR* gene detected by using a [γ³²P] probe complementary to the end of the *aggR* gene sense strand. **b** 5 S rRNA was used as a loading control. **c** Picture of a representative experiment showing the hyper aggregation of the *aggR* + *FRT3′UTR* and the Δ*pnp* strains at 37 °C. The experiment was repeated three times, and a representative experiment is shown.

among certain genes coding flagellar components (such as *lfiE* and *fliS*) and genes associated with the conjugative transfer system (different genes belonging to the *tra* and *trb* operons). Considering that AggR also influences the expression of the AaR protein which, in turn, affects the expression of housekeeping genes[16,17], we studied whether the observed upregulation of metabolic genes in the 042 *aggR* + *FRT3′UTR* strain was detected in an *aar* derivative. Strain 042 *aggR* + *FRT3′UTR aar* exhibited AggR-dependent AaR-independent phenotypes such as enhanced cellular aggregation. In contrast, overexpression of the metabolic genes was not further detected (Supplementary Fig. 4a).

**The *E. coli* 042 *aggR* + *FRT3′UTR* strain shows higher motility and a higher transfer rate of the pAA2 plasmid than the wt 042 strain**. To confirm some of the transcriptomic data, we analyzed both the motility of the 042 wt and 042 *aggR* + *FRT3′UTR* strains, as well as the conjugative transfer of the pAA2 plasmid. The motility of strains 042 wt, 042 pAA2-, 042 Δ*aggR* and 042 *aggR* + *FRT3′UTR* was determined by growing cells on motility plates (LB + agar 0.3% w/v) (Fig. 9a). The results obtained show that the 042 *aggR* + *FRT3′UTR* strain exhibited increased motility compared to that of the wild-type strain (Fig. 9b).

We also compared the conjugative transfer of the pAA2 plasmid in the 042 wt and 042 *aggR* + *FRT3′UTR* strains at both 37 °C and 25 °C. For the wt donor strain, we selected strain 042 *aap-FLAG* (Km^r). Insertions in the *aap* gene are not expected to affect pAA2 conjugation, and the strain has a Km^r resistance marker. The recipient strain was a rifampicin-resistant derivative of the *E. coli* 5 K strain. Hence, the transconjugants were selected in solid medium containing rifampicin and kanamycin. The results (Fig. 9c) showed that the 042 *aggR* + *FRT3′UTR* strain transfers the pAA2 plasmid at a higher frequency than the strain harboring the wt *aggR* gene. It was also apparent that, in both strains, the pAA2 plasmid conjugation frequency was significantly higher at 37 °C than it was at 25 °C.

**The 042 *aggR* + *FRT3′UTR* strain shows higher virulence than the wt strain both in vitro and in vivo**. We also addressed the question of whether the observed expression of the AggR protein in the 042 *aggR* + *FRT3′UTR* strain was correlated with virulence properties in addition to those of the wt strain and whether these hypothetical alterations of virulence may be modified when certain metabolic pathways whose expression is increased in the 042 *aggR* + *FRT3′UTR* strain are inactivated. To that end, we constructed a mutant derivative of the 042 *aggR* + *FRT3′UTR* strain that lacks both the *ast* and *fadAB* pathways (strain 042 *aggR* + *FRT3′UTR ast fad*). The virulence of these strains was assayed both in vitro and in vivo.

In vitro infection experiments were performed with the HEK-Blue™ hTLR4 cell line (InvivoGen). The stimulation of TLR4 was monitored upon infection with cells from the 042 wt, 042 *aggR* + *FRT3′UTR* and 042 *aggR* + *FRT3′UTR ast fad* strains. When compared to the wt strain, TLR4-induced NF-κB activation was significantly increased in the 042 *aggR* + *FRT3′ UTR* strain. Interestingly, depletion of the *ast* and *fadAB* pathways in this later strain resulted in a reduced induction of TLR4 reporter gene expression (Fig. 10a).

In vivo experiments were performed with male C57BL/6 mice, which were administered intragastrical with the strains 042 wt, 042 *aggR* + *FRT3′UTR* and 042 *aggR* + *FRT3′UTR ast fad*, and cytokine expression was measured. Five days post infection (dpi), the mice administered the different strains showed increased expression of cytokines *Il-6* and *Il-1*β in mesenteric lymph node (MLN) leukocytes ($p < 0.05$; Fig. 10b, c). Wild-type bacteria induced a lower expression of *Il-6* ($p < 0.05$) but induced a similar expression of *Il-1*β compared to the 042 *aggR* + *FRT3′UTR* strain. Interestingly, the 042 *aggR* + *FRT3′UTR ast fad* strain showed a significantly lower expression of *Il-1*β ($p < 0.05$). With respect to *Il-6* expression, a *p* value of 0.091 was obtained, which also suggests a tendency toward lower expression.

## Discussion

In this paper, we report novel features of virulence regulation in EAEC strain 042. Alterations in the 3′UTR of the *aggR* gene resulted in increased *aggR* transcription levels, which in turn led to the induction of AggR-regulated genes, such as the major subunit of AAF/II fimbriae, the AafA protein (hence, aggregation of the bacterial cells increased) or Aap dispersin.

The Northern blot analysis of the transcription of the *aggR* gene in the 042 wt and 042 *aggR* + *FRT3′UTR* strains not only confirmed that disruption of the *aggR* 3′UTR increases *aggR* transcript levels but also showed that the *aggR* transcript is unexpectedly large. The in silico analysis of the sequences located 3′ downstream of the *aggR* gene in different pAA plasmids showed that have similar features. A 44-nt sequence located downstream of the *aggR* ORF is shared by the different *aggR* determinants analyzed, and hence, this sequence is thought to be the 3′UTR of the *aggR* gene. A Rho-independent terminator is not present in this 3′UTR. Different IS sequences are inserted downstream of the 3′UTR. These IS elements can differ in different pAA plasmids, but in a common feature their presence increases the distance between the *aggR* ORF and the next Rho-independent terminator. On the other hand, these IS elements provide Rho-dependent terminators. This analysis suggests that random IS elements inserted downstream of the *aggR* gene in the different pAA plasmids can be positively selected because they foster a posttranscriptional regulatory mechanism of the *aggR* gene. Hence, ISs insertions may also influence expression of upstream genes in bacterial genomes. The fact that alterations of the 3′UTR region of the EAEC strain 55989 result in a phenotype similar to that observed in strain 042 *aggR* + *FRT3′UTR* support

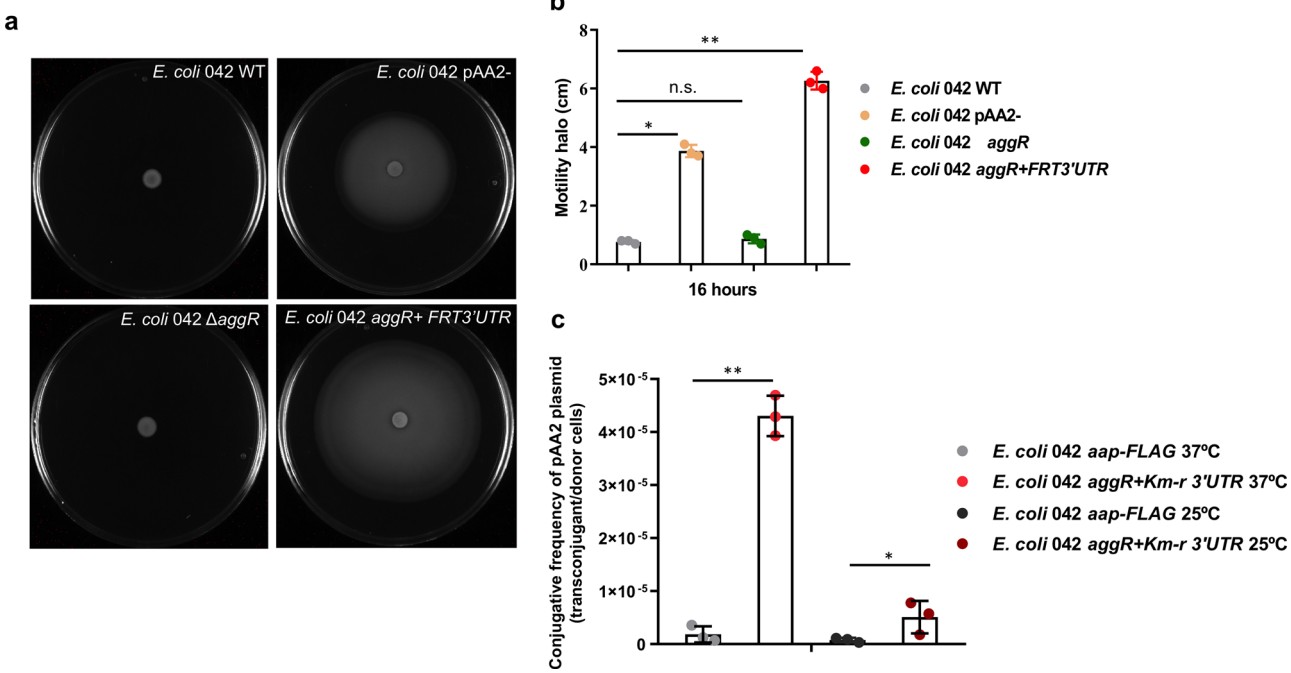

**Fig. 9 Motility and conjugation frequencies quantification of the *E. coli* 042 and derivatives. a** The *aggR + FRT3′UTR* strain is hypermotile. Motility assays of *E. coli* 042 wt, pAA2-, Δ*aggR* and *aggR + FRT3′UTR* were performed at 37 °C. Representative experiment showing the motility halos of the different strains analyzed after incubation on LB 0.3% agar plates for 16 h at 37 °C. **b** Mean diameter of the motility halo in the different strains corresponding to three independent biological replicates. Significance was analyzed by unpaired two–sided Student's t-test. Significance is indicated as *$p < 0.05$ and **$p < 0.01$; n.s: not significant. **c** Conjugation frequencies of the pAA2 plasmid in the 042 *aap-FLAG* and 042 *aggR + FRT3′UTR* donor strains at 37 °C and 25 °C. The 042 *aap-FLAG* strain was used instead of the wt donor strain, as the Km$^r$ determinant associated with the *aap-FLAG* construct presumably does not influence pAA2 plasmid conjugation. Conjugation frequencies correspond to a mean of three independent experiments. Significance is indicated as *$p < 0.05$ and **$p < 0.01$.

the hypothesis that the 3′UTR region plays similar regulatory roles not only in strain 042, but also in other EAEC strains.

The likely consequence of the IS insertions downstream of the *aggR* ORF is the extension of *aggR* transcripts when Rho is not readily available, this resulting in reduced transcript stability. This assumption is supported by several facts, such as the large size of the *aggR* transcripts, the walking RT-PCR data obtained and the observation that the deletion of sequences downstream of the *aggR* gene resulted in increased levels of the *aggR* transcript. Transcriptional read-throughs are not rare events. Up to 34% of known operons from Regulon DB are extended by at least one gene[39].

The roles of long 3′UTRs in the stability, translation and/or localization of eukaryotic mRNAs are well documented[40–43]. The classical view is that the main role of prokaryotic 3′UTRs is harboring a transcriptional terminator that contributes to RNA stabilization, preventing degradation by exonucleases[44–47], has been revised in recent years. Bacterial 3′UTRs can participate in several processes, including the generation and/or interactions with sRNAs, interactions with 5′UTRs and a target of cellular RNases for initiating transcript decay (as reviewed by[47]). In recent years, different reports have provided evidence for the role of 3′UTR in bacterial mRNA decay[34–37]. The length of these 3′UTRs ranges from 63 nt for the *aceA* gene of *C. glutamicum*[35] to 310 nt for the *Salmonella hilD* 3′UTR[36]. In *Yersinia*, long UTRs have been identified in both *Y. pestis*[34] and *Y. enterocolitica*[48]. In the latter species, the existence of Rho-dependent terminators and the participation of PNPase in the decay of transcripts are common features[34,48]. Our findings are very consistent with the results reported for *Yersinia*, although the length of the transcribed 3′UTR may be longer in the *aggR* gene. Our results

suggest that mobile element insertions between the last coding codon of a gene and the corresponding Rho-independent transcriptional terminator may alter gene regulation by generating 3′ extended transcripts that may be susceptible to degradation by cellular ribonucleases. It can be hypothesized that the availability of Rho generates shorter transcripts that are less likely to undergo RNase degradation.

Rho-dependent transcription termination accounts for approximately one-half of the transcription events in *E. coli*[49]. It can occur both inside the coding sequence of genes and at the 3′ end of protein-coding genes (as reviewed in[50,51]). Although it is believed that the latter process is not significantly regulated[50], the recent finding of a sRNA, SraL, that regulates Rho expression[52] suggests that bacterial cells can adjust transcription termination under some conditions by adjusting Rho availability. Rho-dependent termination may be a response to stress conditions[53].

Why does the disruption of 3′UTR *aggR* sequences interfere with the degradation of the *aggR* transcript? The likely answer is that the transcripts of these 3′UTR sequences have evolved to be optimal targets for PNPase, and structural alterations in these sequences thus alter the ability of PNPase to degrade them. Expression of PNPase is autoregulated, and also sensitive to global regulators such as CsrA[54]. Hence, it can not be ruled out that stress conditions that result in altered PNPase levels influence also AggR expression.

Maintaining low levels of AggR expression under noninvasive conditions must be critical for EAEC fitness. The 042 *aggR + FRT3′UTR* strain shows a much lower growth rate than the parental strain. Uncontrolled AggR expression not only leads to the unnecessary expression of virulence factors but also increases cell motility, plasmid pAA conjugation and the synthesis of

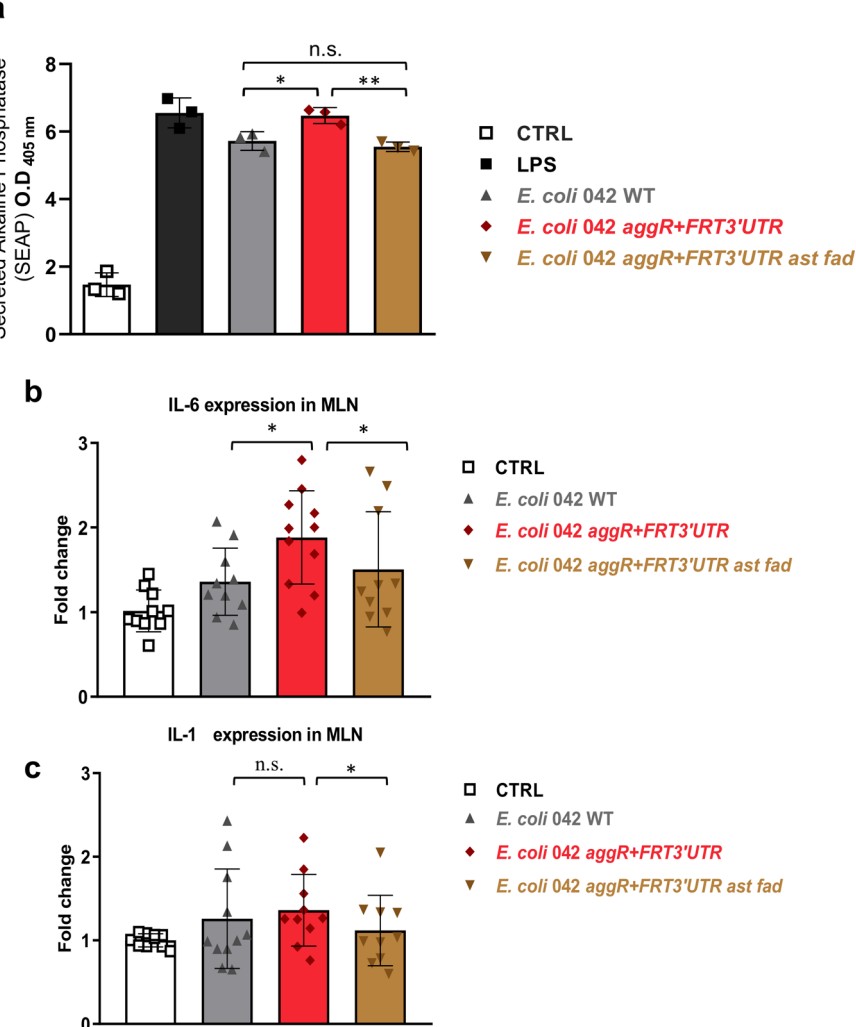

**Fig. 10 In vitro and in vivo quantification of strain 042 *aggR* + *FRT3′UTR* virulence. a** TLR4 mediated NF-κB induction in HEK-Blue[TM] hTLR4 cells upon infection with strains 042 wt, 042 *aggR* + *FRT3′UTR* and 042 *aggR* + *FRT3′UTR ast fad*. Phosphatase alkaline activity was used to measure TLR4-mediated NF-κB induction. CTRL, negative control (culture medium added). LPS, positive control (lipopolysaccharide added). Data are expressed as means values ± SD. Significance is indicated as *$p < 0.05$ and **$p < 0.01$; n.s: not significant. Relative expression of Il-6 (**b**) and Il-1β (**c**) in mesenteric lymph node (MLN) leukocytes of mice infected with the 042 wt strain and *aggR* + *FRT3′UTR* and *aggR* + *FRT3′UTR ast fad* mutants. Mice were intragastrical administered with PBS (control group) and strains 042 wt, *aggR* + *FRT3′UTR* and *aggR* + *FRT3′UTR ast fad*. The results are expressed as the means ± SEM ($n = 10$ mice per group). Significance is indicated as *$p < 0.05$; n.s: not significant.

several metabolic enzymes. The posttranscriptional control of *aggR* transcripts that escape the negative control of regulators such as H-NS may prevent the unnecessary expression of aggR under non-invasive conditions.

Several AggR-regulated genes have been identified in two previous studies. In these studies, the transcriptome of the wt 042 strain was compared with that of an *aggR* isogenic derivative[13,24]. In this study, we took advantage of the "over-expressed AggR" phenotype to further study the AggR regulon. Several of the identified genes that, when compared with the wt strain, are overexpressed in the 042 *aggR* + *FRT3′UTR* strain correspond to the previously reported downregulated genes identified in the 042 Δ*aggR* derivative[13,24], thus validating our analysis. Nevertheless, we have also identified several unreported genes that are directly or indirectly regulated by AggR. Some of these genes are related to flagellar movement and conjugation. It is possible to correlate upregulation of the expression of these genes with the observed higher cell motility and pAA2 conjugation frequency of the 042 *aggR* + *FRT3′UTR* strain compared to the wt strain. These results strongly suggest that, in vivo, AggR

induction not only results in the expression of the previously identified virulence factors but also in increased cell motility and transmission of the virulence plasmid, which may indicate that other plasmid-free *E. coli* cells residing in the intestine may acquire new virulence features.

The upregulated genes in the 042 *aggR* + *FRT3′UTR* strain include several metabolic genes. A careful analysis showed that they correspond to complete metabolic pathways, in which specific metabolic activities are likely increased in cells over-expressing AggR: increased degradation of arginine, fatty acids and GABA. *E. coli* metabolic activities in the intestinal tract have attracted increased research interest in recent years[55–57]. The metabolism of pathogenic *E. coli* strains in the intestine has also been studied[56,58–63]. Metabolites can influence virulence factor expression. Amino acid and fatty acid metabolism modulates the expression of the *E. coli* LEE pathogenicity island in the intestine[63]. Human microbiome metabolites such as hexanoic and heptanoic acids potentiate EAEC-mediated epithelial injury[60]. Microbiota and EHEC-encoded proteases cleave and modulate the activity of type III secretion systems in these pathogens[62]. A

novel finding in this work suggests that the expression of regulators of virulence determinants in response to environmental stimuli can also alter the expression of specific metabolic pathways in the pathogen. Altered pathogen metabolism can interfere with host homeostasis of specific tissues, such as the gut epithelium.

The amino acid L-arginine is a central intestinal metabolite, both as a constituent of protein synthesis and as a regulatory molecule limiting intestinal alterations and maintaining immunophysiological functions[64,65]. It has been established that this amino acid is an integral part of the host defense during infection[66]. The *ast* pathway is used by *E. coli* to grow by leveraging arginine as a carbon source and under conditions of nitrogen limitation[67]. AggR-dependent induction of this pathway in EAEC strains may alter arginine homeostasis in the gut.

The use of fatty acids as carbon source by *E. coli* growth requires the coordinated synthesis of β-oxidation enzymes and a transport system for the fatty acids. At least five separate operons under the control of the FadR repressor are involved in fatty acid availability[68]. Although organic acids shorter than 12 carbon atoms cannot serve as carbon sources for wild-type strains, they can be used by *fadR* mutants, which express constitutively the whole *fad* regulon due to the loss of a repressor protein. Hence, long-chain fatty acid-independent expression of *fad* genes can lead to the catabolism of short-chain fatty acids, and this can happen when AggR expression is induced in strain 042 in the gut. The impact of short-chain fatty acids (SCFAs) on human metabolism has been established (as reviewed in[69]). Depletion of short fatty acids in the gut may also alter gut homeostasis.

The mechanism by which AggR induction results in the increased expression of the metabolic pathways referred above remains to be elucidated. A preliminary analysis performed has shown that the AggR-binding site[13] is not present in the regulatory regions of these metabolic operons. In addition, they are not overexpressed in an *aar* mutant. Taking into account that AggR also influences the expression of the Aar protein which, in turn, affects the expression of housekeeping genes including the nucleotide-associated protein H-NS[16,17], it cannot be ruled out that the Aar regulatory cascade accounts for the induction of the identified metabolic pathways. Our findings are consistent with the identified effect of the AggR/Aar proteins on the lipid metabolism of strain 042[70].

Both in vitro and in vivo experiments correlated increased AggR expression with virulence. The insertion of a FRT DNA in the 3′UTR of *aggR* gene increases the virulence of *E. coli* 042, as shown by the increased expression in infected mice of proinflammatory cytokines, such as *Il-6* in mesenteric lymph nodes, which indicates an intestinal inflammatory response[71]. As well, TLR4 activation in the HEK-Blue™ hTLR4 cell line is higher in the 042 *aggR* + 3′UTR strain than in the 042 wt strain. Interestingly, inactivation of *ast* and *fad* metabolic pathways results in a reduction in the virulence traits of the 042 *aggR* + FRT3′UTR strain, as shown both by the in vitro and in vivo experiments. In vitro, deletion of the *ast* and *fadAB* pathways results in a reduced TLR4 induced NF-κB activation. In vivo, deletion of these metabolic pathways causes a reduction in the expression of *Il-1*β.

The correlation that we report in this study between AggR induction, and the overexpression of specific metabolic pathways may provide new perspectives on the virulence mechanisms of EAEC. A recent genomic analysis of epidemiological EAEC has highlighted the correlation between EAEC and AggR[12]. Indeed, AggR may activate the expression of a strain-dependent range of virulence factors, but also of specific metabolic pathways that definitively contribute to EAEC virulence. These metabolic pathways could correspond to the novel virulence factors under AggR control which existence was recently postulated[72].

## Methods

**Bacterial strains, plasmids and growth conditions**. All the bacterial strains and plasmids used in this work are listed in Supplementary Table 3. The oligonucleotides used in this work are listed in Supplementary Table 4.

Bacterial cultures were grown either in Luria broth (LB) medium (tryptone 10 g/l, yeast extract 5 g/l and sodium chloride 10 g/l), DMEM (supplemented with 0.45% glucose) or M9 minimal medium (supplemented with 0.4% glucose) at 25 °C or 37 °C with vigorous shaking at 200 rpm (*Innova 3100 water bath shaker*, *New Brunswick Scientific*). Liquid cultures were inoculated with a 1:100 dilution of cells cultured overnight (16 h at 37 °C) in LB with vigorous shaking at 200 rpm.

When required, the medium was supplemented with the antibiotics carbenicillin (Cb) at 50 μg/ml, kanamycin (Km) at 50 μg/ml, chloramphenicol (Cm) at 25 μg/ml, or tetracycline (Tc) at 12.5 μg/ml.

**Genetic manipulations**. Genetic insertions in the chromosome of the 042 and 55989 enteroaggregative *E. coli* strains were performed by following the λ-Red recombination protocol[73]. When necessary, the antibiotic resistance determinant was eliminated by using the FLP/FRT-mediated site-specific recombination method[74].

To introduce plasmids into *E. coli*, cells were grown until an optical density at 600 nm (O.D.$_{600}$) of 0.6–0.8 was reached. Then, they were washed several times with 10% glycerol at 4 °C, and the respective plasmids were introduced by electroporation using an Eppendorf gene pulser (Electroporator 2510).

**SDS-PAGE and Western blotting**. Protein extracts were prepared in Laemmli buffer[75] (5% glycerol, 2.5% β-mercaptoethanol, 1.15% SDS, Tris-HCl 31 mM pH 6.6 and 0.05% bromophenol blue). Protein samples were analyzed in 16.5% polyacrylamide Tris-tricine-SDS triphasic gels. After electrophoresis, the proteins were transferred from the gels to PVDF membranes using a semidry electrophoretic transfer cell (*Bio-Rad*) at 15 V for 40 min. For the Western blot analysis, a monoclonal antibody directed against the Flag epitope (*Sigma-Aldrich*) diluted 1:10,000 in a solution of PBS, 0.2% Triton and 3% skim milk was used. The membranes containing the proteins were incubated overnight with the diluted antibody at 4 °C. After incubation, the membranes were washed three times for 10 min with a solution of PBS and 0.2% Triton X-100 (*Sigma-Aldrich*). Thereafter, the membranes were incubated with horseradish peroxidase-conjugated goat anti-mouse IgG (*Promega*) diluted 1:2500 in a solution of PBS and 0.2% Triton X-100 for 45 min at room temperature. After incubation, three additional washing steps with PBS and 0.2% Triton solution were performed for 30 min each time. The immunodetection of the specific protein was performed by enhanced chemiluminescence using a *Molecular Imager ChemiDoc XRS* system and *Quantity One* software (*Bio-Rad*).

**β-Galactosidase assays**. β-Galactosidase activity measurements were performed as described[76]. Values are given as Miller units.

**Cell-free supernatant (secretome)**. Cell-free supernatants were prepared from cultures grown at 37 °C until the O.D.$_{600\ nm}$ of 2.0. For each strain, ten milliliters of cultures were centrifuged, and supernatants were filtered through a 0.22 μm filters, then the total 10 ml of cell-free supernatants containing secreted proteins were precipitated using trichloroacetic acid at a final concentration of 10%. Samples were incubated on ice for 45 min, and then centrifuged for 30 min at 13.400 rpm at room temperature. The pellets were washed once with cold acetone and again centrifuged for 30 min at 13.400 rpm at room temperature. Proteins were solubilized with 1x Laemmli Buffer, boiled for 10 min and loaded into a SDS-PAGE.

**Cell fractionation**. Cell fractionation was prepared as indicated[77]. We used 1 ml of bacterial cells from a culture grown at 37 °C entering the stationary phase O.D.$_{600\ nm}$ of 2.0 for fractionation.

**Protein identification (LC-MS/MS)**. Protein identification was performed at Proteomic Platform (Barcelona Science Park, Barcelona, Spain). Proteins band was cropped from SDS-PAGE gels and processed using the Proteomic Platform standard protocol[78].

**RNA-seq**. RNA extraction, DNase treatment, and the evaluation of RNA quality and cDNA libraries for Illumina sequencing were performed by Vertis Biotechnologie AG, Freising–Weihenstephan, Germany, as described[79].

**Northern blotting**. Northern blot experiments were performed as described[78] with some modifications. The DNA probes used (Supplementary Table 4) were radiolabeled with [γ-$^{32}$P] ATP (PerkinElmer) using T4 polynucleotide kinase (Thermo Scientific) according to the manufacturer's instructions. The radiolabeled probes were then purified using *Sephadex G-25* columns (*GE Healthcare*) following the instructions indicated by the manufacturer and used to detect *aggR* mRNA and the loading control gene *rrfD* (5 S ribosomal RNA).

**Cell aggregation assay**. Cell aggregation assays were performed as described[80] with some modifications. The corresponding cultures were grown in LB medium for 16 h at 25 °C or 37 °C with constant agitation (200 rpm). Then, cells were adjusted to an $OD_{600}$ of 2.0 in 10 ml of culture and transferred to a test tube and maintained in a static state. 100 µl aliquots of the culture surface was removed every 20 min to measure the $OD_{600}$ of the samples. Cell aggregation was represented as a decrease in the absorbance at 600 nm as a consequence of the sedimentation of the cells to the base of the tube. The values are the average of three different experiments, and the standard deviations are shown.

**Motility assay**. The motility assay was performed on tryptone broth (TB) plates (1% tryptone, 0.5% NaCl) containing 0.35% agar. Overnight bacterial cultures grown in LB at 37 °C were spotted (5 µl) on the center of the plates and incubated for 7 h at 37 °C. The experiments were repeated three times with three plates of each strain in each experiment. The colony diameter was measured and plotted, and standard errors were calculated.

**Walking RT-PCR**. RNA for walking RT-PCR was extracted from 5 ml of bacterial cells from an $OD_{600 \ nm}$ of 2.0 grown at 37 °C. Cells were collected with addition of 0.2 volume of stop solution (95% ethanol, 5% phenol), samples were agitated for 30 s and centrifuged 10 min at 7.500 rpm. Bacterial pellets were immediately used for RNA extraction or maintained at −80 °C until use. RNA extraction was performed using Tripure Isolation Reagent (Roche) following manufacture's instructions. Potential traces of DNA were removed by digestion with DNase I (Turbo DNA-free, Ambion), according to the manufacturer's instructions. RNA concentration and RNA quality were measured using a NanoDrop 1000 (Thermo Fisher Scientific). Walking RT-PCR was performed according to[78]. Primers used are listed in Supplementary Table 4.

**Plasmid conjugation**. Experiments performed to quantify the conjugation rate of plasmid pAA2 from E. coli strain 042 were performed in 2xYT liquid medium (16 g/L tryptone, 10 g/L yeast extract and 5 g/L NaCl). First, 900 µl of fresh 2xYT culture medium was tempered at the temperature at which the conjugation experiments were to be performed (25 °C or 37 °C). Next, 50 µl of the plasmid donor strain that had been cultured overnight was added to fresh medium and incubated with continuous shaking for 30 min (for experiments with a conjugation temperature of 37 °C) or 60 min (for experiments with a conjugation temperature of 25 °C). After this incubation time, 200 µl of the culture of the recipient strain was added, and the mixture was incubated at the corresponding temperature under static conditions to facilitate the contact between the donor and recipient cells. Finally, the conjugation was stopped by vortexing the suspension and incubating the cells on ice for 1 min. The mixtures were serially diluted and then plated on LB agar plates containing the corresponding antibiotic. The mating frequency was calculated as the number of transconjugants per donor cell.

**In silico analysis of the DNA sequences**. A bioinformatic analysis performed to identify the presence of possible transcriptional terminators located downstream of the coding region of the aggR gene was performed by using RhoTermPredict[81] software for analyzing the presence of Rho-dependent transcriptional terminators and the Arnold[82] software for searching for the presence of Rho-independent transcription terminators.

**In vivo infection of mice**. Male C57BL/6 mice were purchased from Envigo (Bresso, Italy) and maintained under stable temperature and humidity conditions with a 12-h light 12-h dark cycle and free access to food and water. All animal experiments were carried out in strict adherence to the Guide for the Care and Use of Laboratory Animals, and all protocols were approved by the Ethics Committee for Animal Experimentation of the University of Barcelona and the Catalan government (Refs. 290/19 and 10969, respectively).

At eight weeks of age, the mice were randomly assigned to different groups of E. coli 042 wt, 042 aggR + FRT3'UTR, 042 aggR + FRT3'UTR ast fad bacterial infections or to the control group and inoculated by oral gavage with $10^8$ bacterial CFU or with sterilized saline with phosphate buffer (PBS). Three hours prior to inoculation with bacteria, all the mice were administered cimetidine (50 mg/kg; Sigma-Aldrich) intraperitoneally (i.p.) to reduce acid secretion and improve bacterial survival[83].

Five days postinfection (dpi), the mice were anesthetized i.p. by injection of ketamine/xylazine (100/10 mg/kg) dissolved in 0.9% saline. The mesenteric lymphoid nodes (MLNs) were removed, and lymphocytes were isolated.

Leukocytes from MLNs were obtained as previously described[84]. Briefly, MLNs were finely minced and incubated in a digestion solution composed of RPMI-1640 (Invitrogen, Carlsbad, CA, USA) with 5% inactivated fetal bovine serum (FBS), 100,000 U/L penicillin, 100 mg/L streptomycin, 10 mM HEPES, 2 nM L-glutamine, and 150 U/mL collagenase (Invitrogen, Carlsbad, CA, USA) at 37 °C in a shaker (Thermomixer Comfort Eppendorf, Heuppauge, NY, USA). MLNs were mechanically disaggregated and passed through a stainless-steel mesh. The cell suspension was centrifuged at $500 \times g$ for 10 min at 4 °C. The pelleted cells were resuspended in PBS-FBS.

*NFkB-TLR4 activation reporter cells by LPS signaling*. HEK-Blue™ hTLR4 cell line (InvivoGen) is a stably transfected reporter cell line designed for studying the stimulation of TLR4 and expressing as reporter gene the secreted alkaline phosphatase gene under the control of NF-κB and AP-1. Cells were routinely maintained in DMEM, 10% FBS, 100 U/ml penicillin and 100 µg/ml streptomycin (BioWest, LabClinics), 100 µg/mL Normocin (Invivogen) and 1X HEK-Blue Selection reagent (Invivogen). Cell morphology was analyzed by phase-contrast microscopy (Olympus CKX41).

For experiments, bacteria were grown overnight at 37 °C in LB media with vigorous shaking (150 rpm), then washed once with PBS and diluted in cell culture medium without antibiotics to achieve $10^3$ cfu/ml. HEK-hTLR4 cells were seeded at 25,000 cells/well in a 96-well plate and grown for 24 h before the experiment, then cell culture medium was replaced with new medium containing stimulants (the E. coli 042 wt strain and the corresponding mutants) and incubated overnight in a cell culture incubator at 37 °C, 5% $CO_2$ in a humidified atmosphere. Negative control (culture media) and positive control (lipopolysaccharide (LPS) from E. coli 055:B5 (Sigma) at 10 ng/mL) were used. At the end of the incubation period, cell culture medium was withdrawn, and cells were lysed in PBS containing 1% Triton, 1 mM PMSF and 1 mM EDTA. Protein content of each well was determined using the Bradford Protein Assay (Biorad). SEAP (secreted alkaline phosphatase) activity in the cell culture supernatant was quantified using p-nitrophenyl phosphate as phosphatase substrate according to the manufacturers' instructions (Thermo Scientific, Ref.: 37620). The yellow-colored reaction products were detected using a microplate reader (Clariostar, BMG Labtech) at 405 nm. SEAP activity was calculated according to the formula SEAP activity = ($A_{405nm}$ Test − $A_{405nm}$ Blank) × Total Assay volume (ml)/millimolar extinction coefficient of p-nitrophenol (18.5) × cell culture supernatant employed (ml) × time (min) and normalized to protein content of each well.

All tests and controls were done in triplicate wells, and the average means±standard deviation (SD) were calculated.

*Real-time PCR analysis for in vivo samples*. Total RNA was extracted with TRIzol reagent (Life Technologies, Carlsbad, CA, USA) following the manufacturer's instructions. RNA extraction and retrotranscription were carried out as previously described[71]. Specific primers for mouse hypoxanthine phosphoribosyl transferase 1 (Hprt1), interleukin (Il)-6 and Il-1β were used as previously published[85]. Product fidelity was confirmed by melt-curve analysis. Each PCR run included duplicates of reverse transcripts for each sample and negative controls (reverse transcript-free samples and RNA-free sample). Quantification of the targeted gene transcripts was performed using Hprt1 gene expression as a reference and was carried out by the $2^{-\Delta\Delta CT}$ method[86]. The results are presented as the means ± SEM.

**Statistics and reproducibility**. For β-Galactosidase and conjugation experiments student's t-test was used to determine significance, and the values were obtained by using GraphPad Prism 8 software. A p value of less than 0.05 was considered significant. For In vivo experiments ANOVA and Student's t-test were used to determine the significant differences in the means. p-values < 0.05 were considered significant. Real-time PCR analysis for in vivo samples statistical analysis was performed with Graph Pad Prism software v 8.0. Grubb's test was performed to determine outliers, and Shapiro-Wilk test was used to check the normality of the data distribution. When comparing three or more groups, ANOVA was used when data were normally distributed; otherwise, the nonparametric Kruskal-Wallis test was performed. Statistical differences were considered significant at p < 0.05.

**Reporting summary**. Further information on research design is available in the Nature Research Reporting Summary linked to this article.

## Data availability

The source data for the graphs in the main figures is available as Supplementary Data 1. The uncropped SDS-PAGE, western blot and northern blot images are presented in the Supplementary Information document and numbered as Supplementary Figs. 5 to 13. The RNA-sequencing reads have been deposited in the Gene Expression Omnibus (GEO) Sequence Read Archive of the National Center for Biotechnology Information (GSE160448) under accession numbers GSM4873501 and GSM4873502. Other source data that support the findings of this study are available in the supplementary materials or from the corresponding author upon request.

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

## Acknowledgements
This work was supported by grants BIO2016-76412-C2-1-R and PID2019-107479RB-I00 (AEI/FEDER, UE) and the Ministerio de Economía, Industria y Competitividad, and CERCA Program/Generalitat de Catalunya to A.J. A. Prieto was the recipient of an FPU fellowship from the Ministerio de Educación, Cultura y Deporte, and M. Bernabeu was the recipient of an FI fellowship from the Generalitat de Catalunya. The technical assistance of S. Aznar is acknowledged. Authors wish to thank Prof. Ulrich Dobrindt (University of Münster) who kindly provided the EAEC 55989 strain.

## Author contributions
A.P., M.B., J.F.S.H., A.P.B., L.M., C.B., C.C. and M.H., designed and carried out experiments, A.P., M.H. and A.J. prepared the manuscript.

## Competing interests
The authors declare no competing interests.
