## [Peer Review File · Communications Biology]

Reviewers' comments:

Reviewer #1 (Remarks to the Author):

Summary

Authors in this study show a new regulatory feature associated with the expression of AggR, the AraC/XylS master transcriptional regulator of virulence in EAEC. The work shows that the 3'UTR sequence downstream of the aggR ORF influences expression of aggR and that the insertion of an IS element in the 3'UTR sequence may have compromised the stability of aggR transcripts, perhaps by displacing Rho-independent terminators in the 3'UTR. Bioinformatic analysis revealed that the insertion of IS elements in the 3'UTR sequence downstream the aggR ORF is a common feature exhibited by other clinically relevant EAEC strains. Authors also indirectly showed that aggR transcripts are susceptible to PNPase-mediated degradation given the high expression of aggR in the PNPase mutant.. As expected, the increased in AggR levels by stabilizing aggR transcripts enhances EAEC virulence as demonstrated by high expression of AggR-regulated genes associated with adherence, motility, conjugation and metabolism. A genome-wide transcriptomic analysis of the EAEC strain exhibiting high expression of AggR (042 aggR+FRT3'UTR), revealed high expression of all known genes under AggR regulation, including newly identified genes associated with metabolic pathways that promote degradation of arginine and fatty acids. These data lead them to the conclusion that structural features of the 3'UTR sequence downstream the aggR ORF influences the virulence of EAEC.

Overall impression of this work

This is a well written manuscript, including appropriate experiments and controls and advances our understanding of gene regulation in bacterial pathogenesis. The most relevant aspect of this article is the discovery of the regulatory role of the 3'UTR sequence and the PNPase activity in the levels of AggR gene expression. There is no doubt about the important role that 3'UTRs play in post-transcriptional regulatory mechanisms in Eukaryotes. However, the role of 3'UTR sequences in bacterial virulence is still in its infancy, hence the relevance of the present work in providing new insights on the influence of these regulatory sequences in bacterial virulence. Nevertheless, although authors provided strong evidence for their conclusions, they did not provide experimental evidence that this is not an isolated genetic event, but a general mechanism of gene regulation of aggR ORFs in EAEC strains, which could draw our attention to other genes with similar characteristics (e.g., those regulators that display IS elements in their 3'UTRs). Authors also failed to discuss in what biological circumstances they expect higher expression of AggR in the wild type EAEC strain, in other words, when greater stabilization of AggR transcripts in EAEC may take place. Will this be dependent on the levels of ribonuclease PNPase activity *In vivo*?. Also, some of the genes up-regulated in the EAEC strain with stabilized 3'UTR do not seem to be regulated by AggR, as discussed below.

Specific comments.

1-Lines 175-200 and figures 5-8. By Bioinformatics, the authors compared the genomic structure of the aggR region in the pAA2 plasmid with the corresponding region in other EAEC strains. They showed that the 44bp nucleotide sequence corresponding to the 3'UTR of the aggR gene is highly conserved between strains. However, different IS element(s) are found inserted in 3'UTR sequences, and not all show high homology with the other IS elements. Nevertheless, deletion of 3'UTR or the IS element in EAEC042 resulted in increased expression of AggR, presumably by favoring stabilization of aggR transcripts. Although authors show that at least the 3'UTR of aggR ORFs is highly conserved in other EAEC strains, they did not provide experimental evidence that this mechanism of aggR gene regulation is conserved between EAEC strains, particularly because the strains have different IS insertions in their 3'UTR regions. This could be demonstrated by deletion of 3'UTR or the IS element in any of the other strains.

2. Figure-10. Why the RNaseE mutant does not show aggR transcripts?. Is this required for the proper expression of AggR?.

3. Figure 12, The title of the figure header is incorrect. Authors did not show the aggR+FRT3'UTR

strain is hyper flagellated, but hyper motile. Also flagella does not seem to be regulated by AggR since the pAA2- strain, lacking the aggR gene exhibits higher motility than the wild type EAEC. A recent study using a similar approach in the same EAEC strain (Molecular Microbiology 2019, 111(2), 534–551) has concluded that differential expression of flagellar genes observed in RNA-seq experiments is likely due to stochastic variation and/or phase variation, rather than direct regulation by AggR. So authors should be cautious on their conclusions. They need to acknowledge this possibility or perform complementation experiments using the 042aagR mutant and 042aagR pAggR complement to emphatically show participation of AggR in the regulation of bacterial motility.

4-Is there any evidence of variability in PNase activity in vivo that may influence AggR expression?

Reviewer #2 (Remarks to the Author):

Comments to the Authors:

Prieto et al. (2021) investigates the effect of altering the DNA downstream of the AggR transcription factor in EAEC strain 042. The Authors show that this leads to increased levels of aggR transcript and results in the initiation of the AggR-mediated response in EAEC 042. By examining the DNA downstream of AggR the Authors predict Rho-dependent and Rho independent transcription terminators and suggest that altering downstream DNA affects termination, leading to higher aggR transcript levels. They also show that PNP ribonuclease is important for aggR transcript degradation. Using RNA-seq the Authors identify the genes upregulated when the AggR response is short-circuited, concluding that genes involved in motility and metabolism are indirectly regulated by AggR. The Authors then examine how increased AggR expression affects conjugation of pAA2, motility, TLR4 signalling and interleukin production in mice. The manuscript is well written and contains some very nice data. My queries and comments are detailed below.

1) The Authors do not detail the sequence of the FRT insert downstream of aggR which is carried in strain 042 aggR+FRT3'UTR. As this is usually about 30bp or so, can the Authors explain why in the RT-PCR (Fig. 9) the PCR products for the 042 FRT strain are so much bigger than wild type 042? Is their FRT sequence larger?

2) In Fig S2. The Authors examine the expression of AafA-FLAG and AggR-FLAG at different temperatures. It is not stated in the legend what medium was used (LB or DMEM). It is interesting that most studies examining triggering of the AggR regulon have opted for using DMEM high glucose medium, as LB does not trigger this response. Throughout, the Authors use LB medium, so does using DMEM make any difference? It is clear that inserting/ deleting DNA downstream of aggR greatly influences AggR levels and deregulates the system. Is the 042 aggR-FLAG similarly deregulated and able to express the AggR regulon in LB? Does the FLAG tag interfere with AggR activity? This would be worth mentioning.

3) In the Northern blot experiment in Figure S3, the aggR transcript is very large, well over the 1.3 kb marker that is used. Do the Authors have any idea how large it really is, as it would extend through aafA and probably to aafD on pAA2? If this is the case, it is unclear why the PCR walking experiment (Fig. 9) does not identify transcript with oligos 1 and 6. Can the Authors explain this?

4) The Authors identify a sequence downstream of aggR that they believe is important for the phenomenon that they observe. However, in Fig 5 it looks like E. coli O104:H4 does not have this sequence suggesting that it is not conserved in all EAEC strains. It may be worth showing the comparison of these sequences in the Supplementary Section.

5) The Authors predict that there are a number of Rho-dependent and Rho-independent terminators downstream of *aggR* in various EAEC strains. However, prediction does not mean that they are functional and involved in transcription termination. It is also perplexing that inserting DNA or deleting DNA, especially in the IS inverted repeats, which do not seem to hit the predicted Rho-dependent terminators, still lead to more *aggR* transcript. Furthermore, if in wild type EAEC 042 these terminators are functional, why is the *aggR* transcript the same size as the strain carrying the FRT insertion? I am convinced from the Authors results that stability of the transcript is important but I would need more convincing that this is due to changes in transcription termination.

Minor point. For Fig. 7 using green and red to identify the different terminators could have issues with colour blind readers and strictly speaking Rho-dependent terminators are not stem loops and a different symbol could be chosen.

6) I think is incredibly interesting that the Authors have managed to examine AggR mediated induction in EAEC 042 in LB (rather than DMEM) and this highlights the fact that the cellular levels of AggR are crucial and there may not be an induction signal as such. The Authors RNA-seq work picked up many of the AggR regulated genes found in other studies, as well as other metabolic and flagella genes. As AggR is presumably over expressed in the FRT3`UTR strain it could be argued that induction of these additional genes is simply a result of massive expression of the EAEC fimbriae and secretion systems. In other words, their appearance in the RNA-seq data could be akin to the effect seen with the overexpression of recombinant proteins as cells experience starvation. Thus, in the wild type situation, this may not be observed. Can the Authors comment on this?

7) For the conjugation experiments the Authors pick an *aafA*-Flag EAEC strain as a control to compare with their *aggR* +Km-r 3'UTR strain and state that "insertions in *aafA* are not expected to affect pAA2 conjugation". How do the Authors know this, as if it does, then the results will be misleading? As *AafA* is the major fimbrial subunit, it is possible that the insertion of a FLAG tag influences cell-cell adhesion and the ability of cells to come close together for efficient conjugation. Does the *aafA*-flag strain still form biofilm to a level that is seen for wild type EAEC 042? A simple control experiment like this would show that tagging *AafA* does not interfere with its function.

8) Title of Fig 12 states that the strain is "hyperflagellated". The Authors simply look at the motility of strains and do not show the occurrence or flagella on the cell surface, therefore, a more suitable title should be used.

9) For the TLR4 (Fig. 14) and mice (Fig. 15) experiments I would have expected that a control with EAEC 042 carrying mutations in *ast* and *fad* genes to be included to see if there is an effect in the wild type background. I am concerned that the reason that an effect is seen for the *ast fad* null strain is simply because the cells are sick and do not grow well. How do these strains grow in comparison to their parental strains?

Reviewer #3 (Remarks to the Author):

Prieto et al.

In this work, the authors describe a novel mechanism that influences AggR expression in EAEC, in which the 3'UTR, *aggR* transcripts extend far beyond the *aggR* ORF. Because of the absence of a Rho-independent terminator in these transcripts, they are prone to PNPase-mediated degradation. Structural alterations in the 3'UTR result in increased *aggR* transcript stability, leading to increased AggR levels. The authors therefore investigated the effect of increased AggR levels on EAEC virulence and in addition to the previously reported they found novel AggR-regulated genes that may play relevant roles in EAEC virulence, such as increased mobility and increased pAA2 conjugation

frequency. Furthermore, among the genes exhibiting increased fold change values, the authors could identify those of metabolic pathways that promote increased degradation of arginine, fatty acids and GABA, respectively. It is a very interesting work and their data support their conclusions. I only have small suggestions to improve the paper:

1. The first part of the Results section, ..."the use of *aggR::lacZ* transcriptional"..., is confusing and should be improved (including the use of wt 042 and wt 042 delta-*lacZ* strain; specificity in ..."these 042 derivatives are rougher", they are several o only 042 *aggR+lacZ*3'UTR?).
2. Lines 239-240, there is no reference to figure or they are data not shown.
3. Line 299, write Wild-type bacteria...
4. Line 303, the p value should be 0.091.
5. The Discussion section is very long (more than 6 pages; lines 305-457) and should be shortened (especially, the discussion on metabolic pathways).
6. Figures could be improved, especially Fig. 2, 3, 9 and 11. The labels are very small.

Rew. 1

1- Lines 175-200 and figures 5-8. By Bioinformatics, the authors compared the genomic structure of the *aggR* region in the pAA2 plasmid with the corresponding region in other EAEC strains. They showed that the 44bp nucleotide sequence corresponding to the 3'UTR of the *aggR* gene is highly conserved between strains. However, different IS element(s) are found inserted in 3'UTR sequences, and not all show high homology with the other IS elements. Nevertheless, deletion of 3'UTR or the IS element in EAEC042 resulted in increased expression of AggR, presumably by favoring stabilization of *aggR* transcripts. Although authors show that at least the 3'UTR of *aggR* ORFs is highly conserved in other EAEC strains, they did not provide experimental evidence that this mechanism of *aggR* gene regulation is conserved between EAEC strains, particularly because the strains have different IS insertions in their 3'UTR regions. This could be demonstrated by deletion of 3'UTR or the IS element in any of the other strains.

This has been addressed in the revised version. Authors would like to point out that the genetic manipulating some of these strains can be a very complex task, because they exhibit multiple antibiotic resistances. Nevertheless, we succeed in manipulating the EAEC strain 55989, and show that the phenotype derived from the insertion of a genetic sequence in the 3'UTR region of the *aggR* gene is the same than that of strain 042. (see Results section, novel subsection "Insertion of an *FRT* sequence in the 3'UTR of the *aggR* gene of strain 55989 shows a phenotype similar to that of strain 042 *aggR*+*FRT*3'UTR").

2- Figure-10. Why the RNaseE mutant does not show *aggR* transcripts? Is this required for the proper expression of AggR?

The objective of the experiment shown in Fig 11 was to address whether loss of either RNase E or PNPase function resulted in increased levels of the *aggR* transcript. Due to the low levels of expression of that transcript in the wt background, it is difficult to ascertain whether the results obtained for the RNase E mutant correspond to wt levels or are reduced, but this was not further investigated.

3- Figure 12, The title of the figure header is incorrect. Authors did not show the *aggR*+*FRT*3'UTR strain is hyper flagellated, but hyper motile. Also flagella does not seem to be regulated by AggR since the pAA2- strain, lacking the *aggR* gene exhibits higher motility than the wild type EAEC. A recent study using a similar approach in the same EAEC strain (Molecular Microbiology 2019, 111(2), 534–551) has concluded that differential expression of flagellar genes observed in RNA-seq experiments is likely due to stochastic variation and/or phase variation, rather than direct regulation by AggR. So authors should be cautious on their conclusions. They need to acknowledge this possibility or perform complementation experiments using the 042aagR mutant and 042aagR pAggR complement to emphatically show participation of AggR in the regulation of bacterial motility. We have changed the figure title.

4- Is there any evidence of variability in PNase activity in vivo that may influence AggR expression?

This is obviously a very interesting question. Available information shows that, in addition to autoregulating its own expression at the translational level, the *pnp* gene is subjected to CsrA-mediated regulation. Comments have been introduced in the discussion section.

Rew. 2

1- The Authors do not detail the sequence of the FRT insert downstream of *aggR* which is carried in strain 042 *aggR*+FRT3'UTR. As this is usually about 30bp or so, can the Authors explain why in the RT-PCR (Fig. 9) the PCR products for the 042 FRT strain are so much bigger than wild type 042? Is their FRT sequence larger?

The FRT sequence inserted downstream the *aggR* gene is 84 bp long, and its specific sequence is:

5'GTGTAGGCTGGAGCTGCTTCGAAGTTCCTATACTTTCTAGAGAATAGGAACTTCGGAATAGGAACTAAGGAGGATATTCATATG3'

This is the same sequence as the sequence reported by (Datsenko and Wanner, 2000), located between Priming site 1 and Priming site 2.

B. Predicted "scar" sequences after FLP-mediated excision of antibiotic resistances

Datsenko, K. A., & Wanner, B. L. (2000). One-step inactivation of chromosomal genes in *Escherichia coli* K-12 using PCR products. *Proceedings of the National Academy of Sciences of the United States of America*, 97(12), 6640–6645.

2- In Fig S2. The Authors examine the expression of AafA-FLAG and AggR-FLAG at different temperatures. It is not stated in the legend what medium was used (LB or DMEM). It is interesting that most studies examining triggering of the AggR regulon have opted for using DMEM high glucose medium, as LB does not trigger this response. Throughout, the Authors use LB medium, so does using DMEM make any difference? It is clear that inserting/ deleting DNA downstream of *aggR* greatly influences AggR levels and deregulates the system. Is the 042 *aggR*-FLAG similarly deregulated and able to express the AggR regulon in LB? Does the FLAG tag interfere with AggR activity? This would be worth mentioning.

The reviewer is right, regarding the effect of the growth medium (either DMEM or LB) on AggR expression. The point is that, in our paper, we used LB in order to assess the role of factors different to the culture medium on AggR expression (i.e., temperature or alterations in the 3'UTR). The legend to Fig. S2 has been modified, indicating that cultures were grown in LB medium. Both in LB and DMEM medium, the effect of temperature is similar. As the reviewer acknowledges, altering the *aggR* 3'UTR alters AggR expression. Nevertheless, the phenotype of

clones harboring an AggR-FLAG construction can not be compared to other constructs, as the FLAG sequence may alter AggR activity (which we have not investigated).

3- In the Northern blot experiment in Figure S3, the *aggR* transcript is very large, well over the 1.3 kb marker that is used. Do the Authors have any idea how large it really is, as it would extend through *aafA* and probably to *aafD* on pAA2? If this is the case, it is unclear why the PCR walking experiment (Fig. 9) does not identify transcript with oligos 1 and 6. Can the Authors explain this?

We suspect that the complete *aggR* transcript was not detected by walking RT-PCR experiments due to the Taq DNA polymerase used. Most likely, random transcription termination events may make it difficult to detect very long transcripts.

4- The Authors identify a sequence downstream of *aggR* that they believe is important for the phenomenon that they observe. However, in Fig 5 it looks like *E. coli* O104:H4 does not have this sequence suggesting that it is not conserved in all EAEC strains. It may be worth showing the comparison of these sequences in the Supplementary Section.

The downstream sequence of the *aggR* gene in the *E. coli* O104:H4 strain also shares the same 44 bp homology with respect to the *E. coli* O42 strain, except for two nucleotides. In the Figure 5, the presence of an IS element is overlapping and hiding the 3'UTR region of the *aggR* gene in the O104:H4 strain, but the 3'UTR region shows strong homology to that of strain O42, as shown in the BLAST analysis performed.

CLUSTAL O(1.2.4) multiple sequence alignment

```
042          AAACATGTTTCATATCATTATTTGAGATTGCTATAAACATATTG      44
O104:H4     AAATATGCTTCATATCATTATTTGAGATTGCTATAAACATATTG      44
*** **
```

5- The Authors predict that there are a number of Rho-dependent and Rho-independent terminators downstream of *aggR* in various EAEC strains. However, prediction does not mean that they are functional and involved in transcription termination. It is also perplexing that inserting DNA or deleting DNA, especially in the IS inverted repeats, which do not seem to hit the predicted Rho-dependent terminators, still lead to more *aggR* transcript. Furthermore, if in wild type EAEC O42 these terminators are functional, why is the *aggR* transcript the same size as the strain carrying the FRT insertion? I am convinced from the Authors results that stability of the transcript is important but I would need more convincing that this is due to changes in transcription termination.

As commented above, we routinely used LB medium to grow the different strains. Under these conditions, it is apparent that there is transcriptional readthrough of the *aggR* gene and, whenever our hypothesis should be correct, no Rho-dependent transcription termination takes place at detectable levels. The FRT sequence accounts for yet unknown alterations of the stability of these long transcripts. The hypothesis of Rho-promoted transcription termination under specific growth conditions is just a hypothesis, based on the bioinformatic analysis of transcriptional terminators 3'downstream the *aggR* gene. We clarify this in the discussion section.

Minor point. For Fig. 7 using green and red to identify the different terminators could have issues with colour blind readers and strictly speaking Rho-dependent terminators are not stem loops and a different symbol could be chosen.

Modified following reviewer suggestion.

6- I think is incredibly interesting that the Authors have managed to examine AggR mediated induction in EAEC 042 in LB (rather than DMEM) and this highlights the fact that the cellular levels of AggR are crucial and there may not be an induction signal as such. The Authors RNA-seq work picked up many of the AggR regulated genes found in other studies, as well as other metabolic and flagella genes. As AggR is presumably over expressed in the FRT3'UTR strain it could be argued that induction of these additional genes is simply a result of massive expression of the EAEC fimbriae and secretion systems. In other words, their appearance in the RNA-seq data could be akin to the effect seen with the overexpression of recombinant proteins as cells experience starvation. Thus, in the wild type situation, this may not be observed. Can the Authors comment on this?

We understand that correlating increased AggR expression levels with the induction of specific metabolic pathways may appear surprising. Nevertheless, we have several arguments to support that this is not a side effect, related to starvation-induced stress as such. Several points: -When analyzing the RNAseq data, we realized that the observed overexpression of metabolic genes was not a random event. Only specific pathways are induced. The role of these pathways in homeostasis of gut microbiota is well known.

-Both *in vitro* and *in vivo* experiments show that deletion of two of these pathways results in reduced virulence of the FRT 3'UTR strain.

-We introduce new data in this revised version: the correlation of the induced expression of some of these genes with the activity of the Aar protein (new Fig. 12) and mention a recent paper that demonstrates that Aar protein induces fatty acid metabolism in strain 042.

All these data point to the fact that AggR induction of virulence in strain 042 also leads, via the Aar protein to significant alterations in the activity of specific metabolic pathways.

7- For the conjugation experiments the Authors pick an aafA-Flag EAEC strain as a control to compare with their aggR +Km-r 3'UTR strain and state that "insertions in aafA are not expected to affect pAA2 conjugation". How do the Authors know this, as if it does, then the results will be misleading? As AafA is the major fimbrial subunit, it is possible that the insertion of a FLAG tag influences cell-cell adhesion and the ability of cells to come close together for efficient conjugation. Does the aafA-flag strain still form biofilm to a level that is seen for wild type EAEC 042? A simple control experiment like this would show that tagging AafA does not interfere with its function.

As the reviewer suggested, we checked the ability of strain *E. coli* 042 AafA-FLAG to form biofilm, and our results show that the capacity of this strain to form biofilm was impaired. We decided then to use a different strain. We checked the ability of the strain 042 Aap-FLAG to form biofilm. The results obtained show that the ability of this strain to form biofilm is similar to that of the WT strain, so we performed the new conjugation experiments using this latter strain as the donor of the pAA2 plasmid. The text and Figure (new Fig. 14) have been accordingly modified. Anyhow, it is worth mentioning that the observed effect of AggR overexpression on the

conjugation frequency of the pAA2 plasmid is similar for both donor strains.

8- Title of Fig 12 states that the strain is “hyperflagellated”. The Authors simply look at the motility of strains and do not show the occurrence or flagella on the cell surface, therefore, a more suitable title should be used.

Modified, following reviewer suggestion.

9- For the TLR4 (Fig. 14) and mice (Fig. 15) experiments I would have expected that a control with EAEC 042 carrying mutations in *ast* and *fad* genes to be included to see if there is an effect in the wild type background. I am concerned that the reason that an effect is seen for the *ast fad* null strain is simply because the cells are sick and do not grow well. How do these strains grow in comparison to their parental strains?

The *E. coli aggR*+FRT3'UTR strain shows a reduced growth rate when compared to the wildtype strain when growing in LB medium. The *ast fad* derivative shows only a slightly reduced growth rate when compared to the *aggR*+FRT3'UTR strain. In addition, for the in vitro experiments, the same numbers of cells of each strain were used.

Rew. 3

1- The first part of the Results section, ...“the use of *aggR*::*lacZ* transcriptional”..., is confusing and should be improved (including the use of wt 042 and wt 042 delta-*lacZ* strain; specificity in ...“these 042 derivatives are rougher”, they are several o only 042 *aggR*+*lacZ*3’UTR?).

Modified according to the reviewer’s suggestion.

2- Lines 239-240, there is no reference to figure or they are data not shown.

We have adapted the new figure 11 adding the results of the cellular aggregation assay of the different strains analyzed.

3- Line 299, write Wild-type bacteria...

Modified following the reviewer’s suggestions.

4- Line 303, the p value should be 0.091.

Modified.

5- The Discussion section is very long (more than 6 pages; lines 305-457) and should be shortened (especially, the discussion on metabolic pathways).

We have reduced the discussion section following the reviewer’s suggestion.

6- Figures could be improved, especially Fig. 2, 3, 9 and 11. The labels are very small.

Modified, following reviewer’s suggestion.

Reviewers' comments:

Reviewer #1 (Remarks to the Author):

Authors satisfactorily demonstrated their previous findings. I have no other comments.

Reviewer #2 (Remarks to the Author):

Prieto et al. (2021) investigates the effect of altering the DNA downstream of the AggR transcription factor in EAEC strain 042 (and EAEC 55989). The Authors show that this leads to increased levels of aggR transcript and results in the initiation of the AggR-mediated response in EAEC 042. The Authors show that PNP ribonuclease is important for aggR transcript degradation and using RNA-seq the Authors identify the genes upregulated when the AggR response is short-circuited, concluding that genes involved in motility and metabolism are indirectly regulated by AggR. The Authors then examine how increased AggR expression affects conjugation of pAA2, motility, TLR4 signalling and interleukin production in mice.

Again, the manuscript is well written and contains some very nice data. As the Authors have answered many of my previous queries, I only have a few minor comments and suggestions detailed below.

- 1) Abstract. Line 42. GABA could do with defining here on first use.
- 2) Results. Line 167. "3' noncoding sequence of the aggR gene". To me it would be more correct to say downstream of the aggR gene.
- 3) Results. Line 183. "The IS1A element of the pAA2 plasmid does not show homology with the other IS elements, which share a high degree of homology." Needs clarifying.
- 4) Results. Line 194. "In contrast, several (potential) Rho-dependent terminators...". These sites are predicted and in general rut sites are not that conserved so there is no proof that these sites will cause loading of Rho onto the aggR transcript. The Discussion also focuses on the involvement of Rho but in my opinion there is really no evidence that Rho is involved in aggR transcript instability.
- 5) Results. Line 213. "As it was shown fort (for) the E. coli 042 strain, strain 55989 aggR+FRT3' UTR also showed the overexpression of the major subunit of the AAF/III determinant (Agg3A and the dispersin (Aap) proteins) when compared to the wt strain." Change "fort" to "for" and consider revising the sentence, as it is a little complicated.
- 6) Results. Line 222. "The first deletion corresponded to a (the) 44 bp 3'UTR established in silico." Change "a" to "the".
- 7) Results. Line 227. Title: "A transcriptional readthrough is formed by the aggR". Consider revising.
- 8) Results. Line 230. "transcripts initiated downstream of (at) the aggR promoter extend far beyond the aggR ORF". I would perhaps say that transcript initiation takes at the promoter rather than "downstream of".
- 9) Figure 6. The -10 hexamer sequence should be extended slightly to CATTCT. Why are there two arrows upstream of aggR? Is there more than one transcription start site?
- 10) Results. Line 253. "Whereas the transcription of the aggR gene was similar in the wt strain and the RNase E mutant". Transcript levels for wild type EAEC 042 and the RNase E mutant are not the same on Figure 11B so this sentence should be rephrased. Reviewer 1 raised this point.
- 11) Legend of Figure 12. Line 983. "RT-qPCR in E. coli 55989 aggR+FRT3'UTR". Should this be 042?
- 12) Results Line 274. The Authors have introduced a new experiment into the manuscript detailing RNA-seq with an aar mutant. Currently it feels that it has been "dropped into" the paper without explanation for the rationale behind this. An introductory sentence or two would definitely help Readers. As the upregulation of metabolic genes was not observed in this strain it is important to know that conventional AggR regulated genes were still induced. This should be commented on and possibly the data included in the supplementary.
- 13) I still feel that a suitable control of EAEC 042 carrying mutations in ast and fad genes should be

included in at least the TLR4 induction experiments (Fig. 15). If wild-type EAEC 042 similarly mounts a reprogramming of its metabolism then you should see the same effect.

1) Abstract. Line 42. GABA could do with defining here on first use.

Done

2) Results. Line 167. "3' noncoding sequence of the *aggR* gene". To me it would be more correct to say downstream of the *aggR* gene.

Done

3) Results. Line 183. "The IS1A element of the pAA2 plasmid does not show homology with the other IS elements, which share a high degree of homology." Needs clarifying.

Done

4) Results. Line 194. "In contrast, several (potential) Rho-dependent terminators...". These sites are predicted and in general rut sites are not that conserved so there is no proof that these sites will cause loading of Rho onto the *aggR* transcript. The Discussion also focuses on the involvement of Rho but in my opinion there is really no evidence that Rho is involved in *aggR* transcript instability.

We find interesting to show that no Rho-independent terminator is placed close to the 3' end of the *aggR* gene. On the other hand, we consider worth mentioning the presence of bioinformatically identified Rho-dependent terminators. We mention in the discussion that the participation of Rho is only hypothetical, but we find that this is an interesting hypothesis that can be supported by future experimental work.

5) Results. Line 213. "As it was shown fort (for) the E. coli 042 strain, strain 55989 *aggR*+FRT3'UTR also showed the overexpression of the major subunit of the AAF/III determinant (Agg3A and the dispersin (Aap) proteins) when compared to the wt strain." Change "fort" to "for" and consider revising the sentence, as it is a little complicated.

Done

6) Results. Line 222. "The first deletion corresponded to a (the) 44 bp 3'UTR established in silico." Change "a" to "the".

Done

7) Results. Line 227. Title: "A transcriptional readthrough is formed by the *aggR*". Consider revising.

Done

8) Results. Line 230. "transcripts initiated downstream of (at) the *aggR* promoter extend far beyond the *aggR* ORF". I would perhaps say that transcript initiation takes at the promoter rather than "downstream of".

Done

9) Figure 6. The -10 hexamer sequence should be extended slightly to CATTCT. Why are there two arrows upstream of *aggR*? Is there more than one transcription start site?

Clarified in the legend to the figure.

10) Results. Line 253. "Whereas the transcription of the *aggR* gene was similar in the wt strain and the RNase E mutant". Transcript levels for wild type EAEC 042 and the RNase E mutant are not the same on Figure 11B so this sentence should be rephrased. Reviewer 1 raised this point. The sentence has been accordingly modified.

11) Legend of Figure 12. Line 983. "RT-qPCR in *E. coli* 55989 *aggR+FRT3'UTR*". Should this be 042?

Done.

12) Results Line 274. The Authors have introduced a new experiment into the manuscript detailing RNA-seq with an *aar* mutant. Currently it feels that it has been "dropped into" the paper without explanation for the rationale behind this. An introductory sentence or two would definitely help Readers. As the upregulation of metabolic genes was not observed in this strain it is important to know that conventional AggR regulated genes were still induced. This should be commented on and possibly the data included in the supplementary.

Introductory sentences have been introduced. In addition, the effect of AggR/AaR proteins on the gene expression profile of EAEC is well characterized (Santiago et al., 2017. Plos Pathogens e1006545). Specific genetic determinants, such as *aaf*, are regulated by AggR and not by Aar. The *aar* mutant derivative of strain 042 *aggR+FRT3'UTR* shows a similar cellular aggregation than the parental 042 *aggR+FRT3'UTR* strain. This is now mentioned in the text.

13) I still feel that a suitable control of EAEC 042 carrying mutations in *ast* and *fad* genes should be included in at least the TLR4 induction experiments (Fig. 15). If wild-type EAEC 042 similarly mounts a reprogramming of its metabolism then you should see the same effect.

We consider that these studies are beyond the scope of our work. Detailed approaches shall be required to provide evidence for the wt strain inducing these metabolic pathways, either in vivo or in vitro. The specific environmental stimuli leading to the overexpression of these metabolic pathways in the wt strain may not be easy to determine. We would also like to point out that the relationship that we have found between the AggR regulator and one of these metabolic pathways (fatty acid metabolism) has also been reported in a recent publication by using a different approach.